# High kinetic inductance cavity arrays for compact band engineering and topology-based disorder meters

Vincent Jouanny [1,2] ✉, Simone Frasca [1,2], Vera Jo Weibel [1,2], léo Peyruchat [1,2], Marco Scigliuzzo [2,3], Fabian Oppliger[1,2], Franco De Palma[1,2], Davide Sbroggiò[1,2], Guillaume Beaulieu [1,2], Oded Zilberberg [4] & Pasquale Scarlino [1,2] ✉

Superconducting microwave metamaterials offer enormous potential for quantum optics and information science, enabling the development of advanced quantum technologies for sensing and amplification. In the context of circuit quantum electrodynamics, such metamaterials can be implemented as coupled cavity arrays (CCAs). In the continuous effort to miniaturize quantum devices for increasing scalability, minimizing the footprint of CCAs while preserving low disorder becomes paramount. In this work, we present a compact CCA architecture using superconducting NbN thin films manifesting high kinetic inductance. The latter enables high-impedance CCA (~1.5 kΩ), while reducing the resonator footprint. We demonstrate its versatility and scalability by engineering one-dimensional CCAs with up to 100 resonators and with structures that exhibit multiple bandgaps. Additionally, we quantitatively investigate disorder in the CCAs using symmetry-protected topological SSH edge modes, from which we extract a resonator frequency scattering of $0.22^{+0.04}_{-0.03}\%$. Our platform opens up exciting prospects for analog quantum simulations of many-body physics with ultrastrongly coupled emitters.

Metamaterials made of superconducting circuits[1] have emerged as highly versatile platforms at the forefront of quantum technologies, offering a broad range of applications encompassing sensing[2], amplification[3], and quantum information processing[4,5]. Moreover, when quantum emitters are strongly or ultrastrongly coupled to microwave superconducting metamaterials that feature high quality and mode density[6–8], or to structured photonic baths[9–11], it provides a valuable framework for exploring many-body phenomena through analog quantum simulation[12–14]. In particular, coupled cavity arrays (CCAs) have emerged as a flexible architecture for realizing artificial photonic materials in the tight-binding limit[15,16], enabling the creation of band structures with varying complexity[17,18]. These structures are even capable of realizing non-Euclidean geometries[19], non-trivial tolopogical lattices[20–22], and flat bands[23], offering insights into complex many-body physics[24] and enabling various quantum information processing tasks[4,5,25].

Conventional on-chip superconducting CCAs are realized with distributed coplanar waveguide (CPW)[26] or lumped element LC[27] resonators, typically made of aluminum (Al) or niobium (Nb) superconducting thin films. This architecture allows for arbitrary band engineering by tailoring the frequency and coupling of the cavities[1,19,28]. Despite this flexibility, the large physical footprint of a single resonator at frequencies ~5 GHz covers several millimeters' square[26,29], and can restrict the scalability of the array. Recent research

[1]Hybrid Quantum Circuits Laboratory (HQC), Institute of Physics, École Polytechnique Fédérale de Lausanne (EPFL), 1015 Lausanne, Switzerland. [2]Center for Quantum Science and Engineering, Institute of Physics, École Polytechnique Fédérale de Lausanne (EPFL), 1015 Lausanne, Switzerland. [3]Laboratory of Photonics and Quantum Measurements (LPQM), Institute of Physics, École Polytechnique Fédérale de Lausanne (EPFL), 1015 Lausanne, Switzerland. [4]Department of Physics, University of Konstanz, D-78457 Konstanz, Germany. ✉e-mail: vincent.jouanny@epfl.ch; pasquale.scarlino@epfl.ch

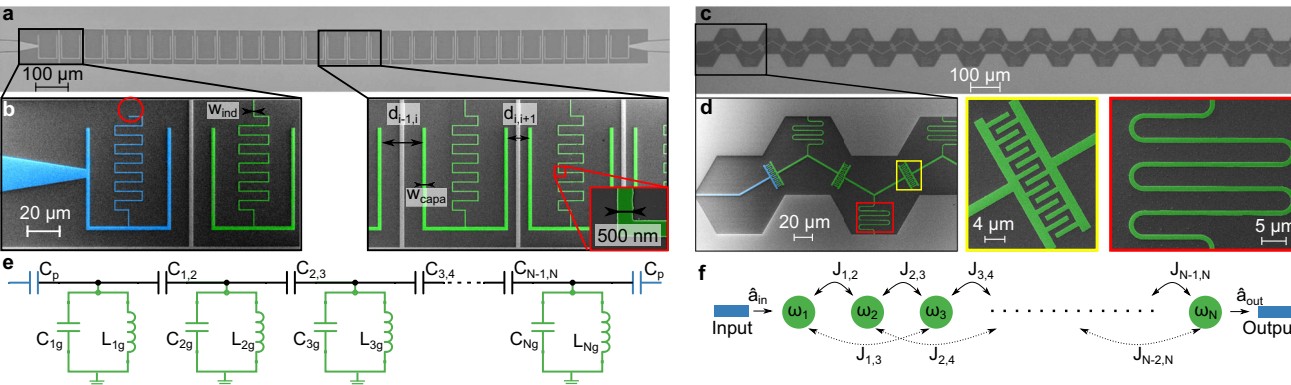

**Fig. 1 | High-impedance CCA. a** Optical micrograph of a representative linear-design CCA comprising 25 resonators with rectangular base unit. The dark part is silicon while the light part is NbN. **b** False-colored scanning electron micrographs of the zoomed-in regions of the CCA [marked by black frames in (**a**)]. The microwave port is colored in blue and the cavities in green. Notice that the coupling port (blue) does not have an inductor shunted to ground. Inset: Further zoom-in on a portion of the inductor [marked by a red frame in (**b**)]. **c** Optical micrograph of a representative zigzag-design CCA comprising 26 resonators with hexagonal base unit. **d** False-colored scanning electron micrograph of the zoomed-in region of the CCA [marked in (**c**)]. The microwave port is colored in blue and the cavities in green. The insets [yellow (red) frame] show a zoom-in on the mutual capacitor between two cavities (the inductor of a cavity). **e** Schematic of the lumped-element model of the CCAs. Each cavity is modeled as an LC resonator with an inductor $L_{ig}$ and a capacitor $C_{ig}$ to ground; the $i$ and $j$ cavities are coupled via a mutual capacitor $C_{i,j}$. The cavities at the edges of the CCA are coupled to the microwave ports via the capacitors $C_p$ in blue. **f** Schematic of the corresponding chain Hamiltonian including first (second) neighbor interaction $J_{i,i+1}$ ($J_{i,i+2}$) between cavities $i$ and $i + 1$ ($i + 2$), cf. Eq. (2). The input and output microwave ports are represented in blue, and $\hat{a}_{in(out)}$ indicates the input (output) field operator.

has explored alternative approaches, such as the replacement of the geometric inductance with the Josephson inductance of compact junction arrays[25,30,31]. However, achieving control of the Josephson junction inductance with an imprecision below a few percent[31,32] remains challenging, leading to significant variability in cavity parameters and impacting the spectral properties of the CCAs. Additionally, this approach introduces significant nonlinearity, affecting the higher excitation manifold of the CCA. Despite the variety of approaches, the challenge of realizing a CCA made of unit cells that combine simultaneously high-quality, ultra compactness, and weak non-linearities, while maintaining a low overhead in fabrication, still remains elusive. To fully harness the potential of CCAs while dramatically reducing their size, it is crucial to maintain low scattering of the cavity frequencies and inter-site coupling, as well as to develop methods to efficiently quantify the impact of disorder.

In this work, we report on a compact and versatile lumped-element CCA architecture characterized by low disorder, with only $0.22^{+0.04}_{-0.03}\%$ deviation in resonator frequency. The resonators are made of high kinetic inductance NbN thin film resulting in compact inductors[33–35]. We show the versatility and scalability of the platform by engineering one-dimensional CCAs with up to 100 resonators with multiple band-structures. To efficiently quantify the amount of disorder in the system, we develop a *topology-inspired* metric for assessing the resonators' frequency scattering by systematically exploring the in-gap mode distribution of CCAs that realize the Su-Schrieffer-Heeger (SSH) chain[36]. Due to the bulk-edge correspondence[37] in-gap modes are also sensitive to chiral symmetry-breaking disorder in the bulk of the CCA. Thus, by focusing solely on the edge modes we can infer the overall disorder present in the entire system. Notably, the high kinetic inductance of our devices enables the realization of high-impedance resonators in the array. This characteristic increases the coupling to charge degree of freedom of both superconducting[38] and semiconducting[39] qubits, enhancing the possibility to achieve the ultra-strong coupling regime[40].

This ultracompact, high-impedance platform significantly enhances the density of modes within a given footprint, without compromising the quality or increasing the disorder. This enables the exploration of complex and high density of photonic states necessary for emulating the effect of highly structured non-Markovian environment of open

quantum system[17,18]. This development lays the groundwork for integrating quantum emitters into our bath-engineered CCA environment.

## Results

### Platform

We design, simulate, fabricate, and investigate 1-dimensional (1D) CCAs comprising rectangular-shaped (Fig. 1a, b) and hexagonal-shaped base units (Fig. 1c, d). Each CCA is fabricated from a high kinetic inductance NbN thin film (see Methods) and can be modeled as an array of $N$ superconducting lumped-element LC resonators, as schematically represented in Fig. 1e. Each resonator is defined by a capacitor with total capacitance $C_{\Sigma,i} \approx C_{ig} + C_{i-1,i} + C_{i,i+1}$, where $C_{ig}$ represents the capacitance to ground of the $i^{th}$ cavity, which also shares mutual capacitances, $C_{i-1,i}$ and $C_{i,i+1}$ with its neighboring resonators. These latter two capacitances can be adjusted by varying the spacing between the resonators $d_{i-1,i}$ and $d_{i,i+1}$, or by adjusting the inter-digitated capacitor, see Fig. 1b, d. The resonance frequency of the $i^{th}$ resonator, denoted as $\omega_i/2\pi$, is determined by $1/\sqrt{L_{ig}C_{\Sigma,i}}$. Both $C_{ig}$ and $C_{i\pm1,i}$ can be tailored independently, while keeping $C_{\Sigma,i}$ and, consequently, the resonance frequency $\omega_i/2\pi$, constant. The inductance to ground, $L_{ig}$, of the nanowire inductor with width $w_{ind}$ and length $l_{ind}$ (Fig. 1b, d) can be expressed as[33,34]

$$L_{ig} = L_k + L_{geo} = L_{k,\square}\frac{l_{ind}}{w_{ind}} + L_{geo}, \tag{1}$$

with $L_k$ ($L_{k,\square}$) representing the (sheet) kinetic inductance, and $L_{geo}$ accounting for the geometric inductance. In our case, the ratio $L_k/L_{geo} \sim 250$ indicates that the inductance of the CCA is completely dominated by its kinetic contribution. Leveraging this property, we are able to engineer resonators with a significantly reduced footprint typically down to $50 \times 75\ \mu m^2$[35]. This size is remarkably smaller if compared to conventional lumped-element resonators (~100 × smaller)[20,41] and CPW distributed resonators (~1500 × smaller)[19,26]. Due to the large kinetic inductance, the resonators in the array present an impedance $Z_i = \sqrt{L_{ig}/C_{\Sigma,i}}$ of approximately 1.5 k$\Omega$ (0.8 k$\Omega$) for the rectangular (hexagonal) geometry. This high impedance enhances the capacitive couplings between resonators ($J_{i,i+1} \propto \sqrt{Z_iZ_{i+1}}$) and to quantum emitters ($g_i \propto \sqrt{Z_i}$).

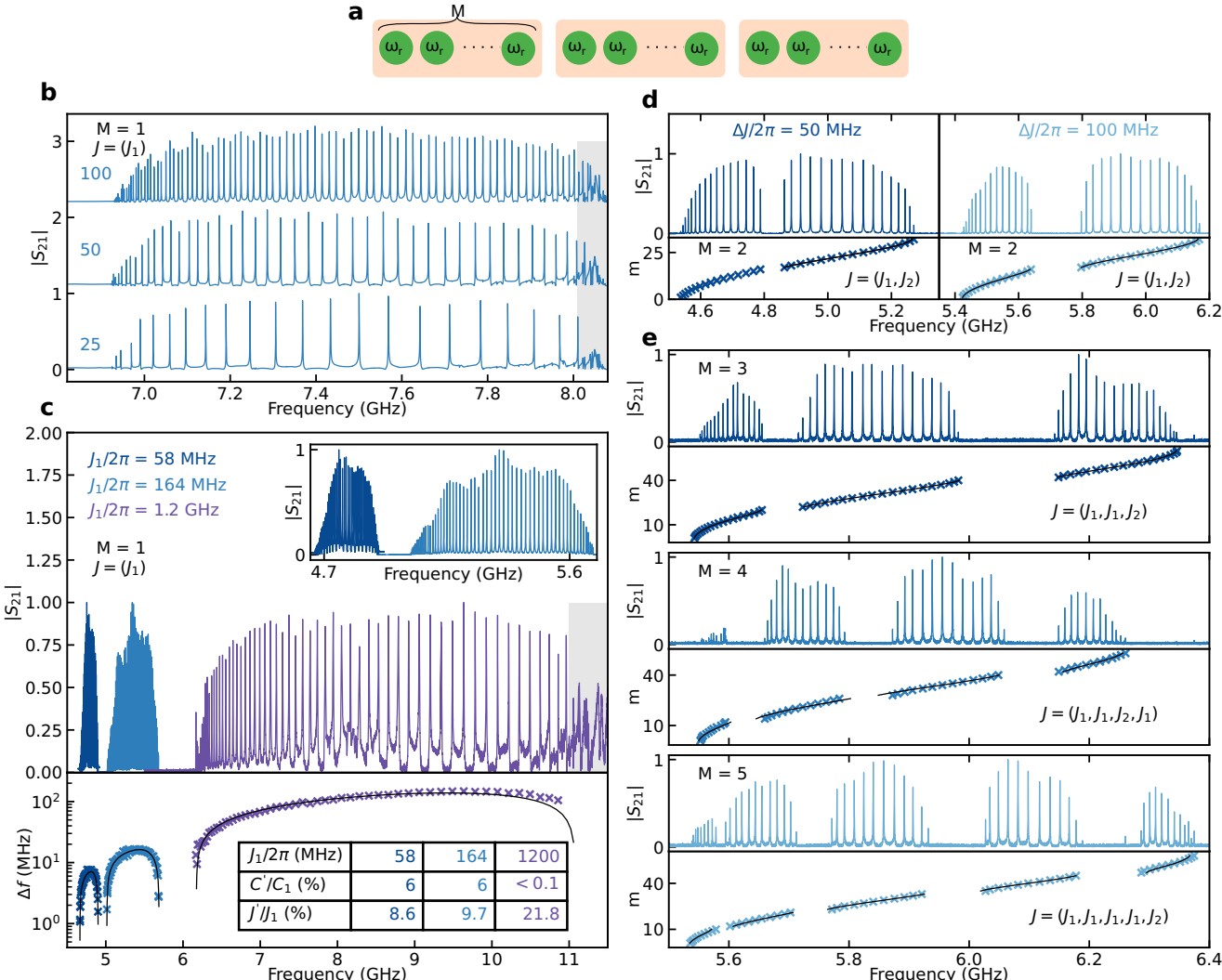

**Fig. 2 | Band engineering. a** General CCA schematic displaying the cavities in green, with a unit cell including $M$ cavities. **b** Transmission spectrum $|S_{21}|$ for CCAs with $N = 25$, 50 and 100 cavities with rectangular design with $M = 1$. The shaded region around 8.05 GHz highlights the presence of microwave chip slot modes. **c** Top: Same as (**b**) for $M = 1$ CCAs with $J_1/2\pi = 58$ MHz (rectangular design), $J_1/2\pi = 164$ MHz (rectangular design), and $J_1/2\pi = 1200$ MHz (hexagonal design). Additional slot modes are visible around 11.1 GHz. Inset: a zoom-in on the two lower coupling CCAs in the main panel. Bottom: Frequency difference, $\Delta f = f_i - f_{i+1}$, between two consecutive modes for the corresponding CCAs. The crosses represent the $\Delta f$ extracted for the CCAs, each plotted in relative to the averaged modes' frequencies $(f_i + f_{i+1})/2$. The continuous black lines indicate fits of the extracted $\Delta f$ according to the eigenmodes of the CCAs Hamiltonian (see Methods). Inset table collects the coupling $J_1/2\pi$, the stray capacitance ratio $C'/C_\Sigma$, where $C' = C_{i,i+2}$, and next nearest-neighbor couplings ratio $J'/J_1$ for the three CCAs. **d** Top: Same as **c** for dimerized CCAs (rectangular design, $M = 2$) with $\Delta J/2\pi = |J_2 - J_1|/2\pi \cdot 50$ MHz (100 MHz) on the left (right). Bottom: Extracted modes (crosses) and their fits to the eigenmodes of the CCAs Hamiltonian (continuous line) (see Methods). **e** Same as **c** for (top to bottom) unit cells with $M = 3$, 4, and 5. All the transmission spectra are normalized by their maximum amplitude.

The Hamiltonian of the system is derived following standard Lagrangian circuit quantization[42] (see Methods), and takes the form

$$\hat{H} = \sum_{i=1}^{N} \omega_i \hat{a}_i^\dagger \hat{a}_i + \sum_{q=1}^{Q} \sum_{i=1}^{N-q} J_{i,i+q}\left(\hat{a}_i^\dagger \hat{a}_{i+q} + h.c.\right), \quad (2)$$

up to the $Q^{\text{th}}$ order in coupling in $\hbar = 1$ units. Here, $\hat{a}_i$ and $\hat{a}_i^\dagger$ are the photonic annihilation and creation operators at site $i$. The nearest-neighbor coupling terms, $J_{i,i+1}$, originate mainly from direct capacitive coupling. In our model, we neglect inductive coupling due to the resonators' high impedance, as it scales as $1/\sqrt{Z_i}$. Coupling terms of order $q > 1$ have a double origin: stray capacitive coupling between next nearest-neighbor resonators and from the inversion of the capacitance matrix in the circuit's Lagrangian, the latter increases with the ratio $C_{i,i+1}/C_{\Sigma,i}$ (see Methods and

Supplementary file). In the following, we focus on CCAs realized with resonators with degenerate frequencies, $\omega_r/2\pi$. To ensure this degeneracy, we introduce driving ports shaped and positioned exactly as a cavity of the array but whose inductor is not shunted to ground. Such ghost ports ensure a uniform capacitive environment for both edge and bulk resonators (see blue false-colored part in Fig. 1b).

## Band engineering

We begin by characterizing the extensibility of the introduced compact CCA platform by measuring chains with homogeneous coupling, i.e., $J_{i,i+1} = J_1$ and $J_{i,i+2} = J'$, cf. Fig. 1d. To this end, we set $C_{ig} = C_g$, $L_{ig} = L_g$ and $C_{i,i+1} = C_1$. Note that we have $M = 1$ resonators per unit cell [cf. Fig. 2a], and for $J_1 \gg J'$, we expect a finite-size sampling of a cosine dispersion, i.e., the emergence of a passband centered around $\omega_r/2\pi$ with a span of $4J_1$[25]. In Fig. 2b, we report on the

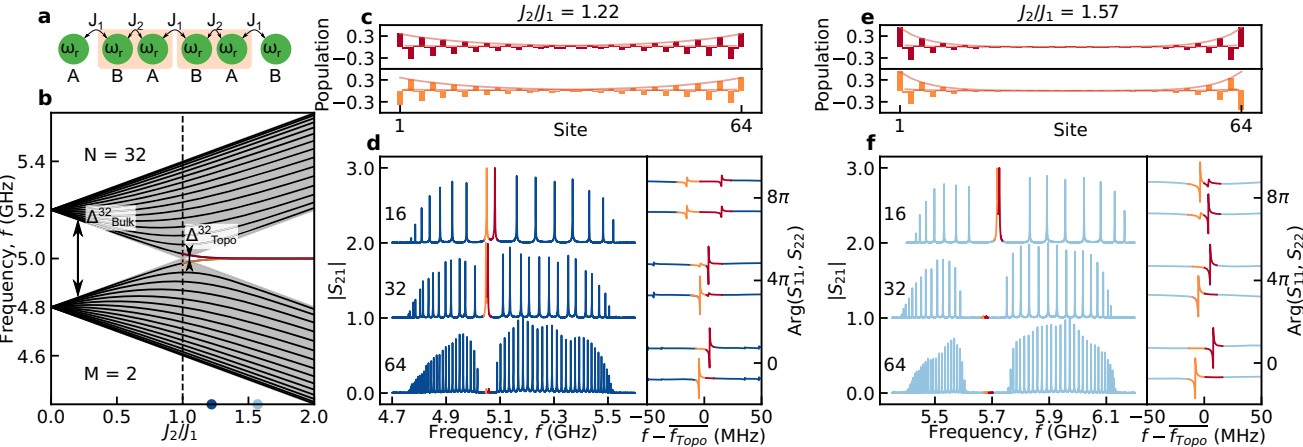

**Fig. 3 | Engineering SSH edge modes. a** Schematic of the SSH model. Each unit cell contains two cavities A and B, both with frequency $\omega_r$. $J_1$ and $J_2$ are respectively the intracell and intercell coupling. **b** Simulated phase transition diagram of the SSH model from trivial ($J_1 > J_2$) to topological ($J_1 < J_2$) phase, with $J' = 0$. The black lines represent the bulk modes for a CCA with $N = 32$. For $J_1 \neq J_2$, the system presents a bandgap of size $\Delta_{\mathrm{Bulk}}^{32}$. In the non-trivial phase, two hybridized SSH edge modes (red and orange) are enabled at the center of the bandgap and are separated by $\Delta_{\mathrm{Topo}}^{32}$. The gray area represents the phase transition diagram $N \to \infty$. **c** (**e**) Simulated photonic population of the CCA with $N = 64$ in correspondence of the symmetric (red) and antisymmetric (orange) hybridized SSH edge modes in the weakly localized configuration, $J_2/J_1 = 1.22$ (strongly localized configuration, $J_2/J_1 = 1.57$) according to the eigenvectors of the CCAs Hamiltonian (see Methods). **d** (**f**) Left: Transmission spectrum $|S_{21}|$, for CCAs with $J_2/J_1 = 1.22$ ($J_2/J_1 = 1.57$) and $N = 16$, 32 and 64. Right: Reflection spectra Arg $S_{11,22}$, as a function of the frequency detuning $f - \overline{f_{\mathrm{Topo}}}$, for a frequency region of 100 MHz around the SSH edge modes. $\overline{f_{\mathrm{Topo}}}$ is the mean frequency of the two SSH edge modes. The modes in red and orange represent the symmetric and antisymmetric hybridized SSH edge modes, respectively.

transmission spectra of several such homogeneous CCAs with $N$ up to 100 sites. The transmission, $|S_{21}|$, is measured in a cryogenic setup at 10 mK with a vector network analyzer (see Supplementary file). Each CCA transmission is normalized with respect to its maximum transmission amplitude. We observe $N$ distinct peaks, corresponding to the eigenmodes of the CCA. The modes at the center of the band have respectively larger coupling to the ports [higher peaks], $\kappa_{\mathrm{ext}}$, and sparser frequency spacing, $\Delta f$, relative to the smaller coupling [lower peaks] and higher mode density at the band edges[25] (Fig. 2b and Supplementary file). In all examined CCAs, we resolve the majority of the modes; for example, we detect as many as ~90 distinct modes in the case of $N = 100$ CCA. The missing modes can be attributed to two dominant factors: (i) the modes at the edges of the pass-band have lower visibility, and (ii) to avoid erroneous counting, we exclude the frequency region around 8.1 GHz (Fig. 2b), where chip slot modes are present. On average, we extract individual mode single-photon internal dissipation rates $\kappa_{\mathrm{int}}^{\mathrm{Mode}}/2\pi$ of 100 kHz for $\omega_r/2\pi = 5$ GHz (internal quality factor $Q_{\mathrm{int}}^{\mathrm{Mode}} \sim 50 \times 10^3$), indicating low-loss CCAs (see Supplementary file). Remarkably, fabricating CCAs with a high number of cavities does not degrade the low-power quality factor of the device[35].

We proceed to demonstrate high control over the inter-site coupling, see Fig. 2c. By redistributing the components contributing to the total capacitance, $C_\Sigma$, of each resonator, specifically adjusting the capacitance to ground, $C_g$, and mutual capacitance, $C_1$, we can modulate the inter-site coupling rate. This allows us to engineer multimode environments with bandwidths ranging from approximately 230 MHz up to 4.8 GHz, resulting in a free spectral range spanning from hundreds of MHz down to 1 MHz. Considering the low-dissipation rates and the possibility to engineer narrow free spectral range, this architecture opens exciting prospects for exploring superstrong light-matter coupling[7] and many-body Hamiltonians in the strongly non-linear regime[43,44]. Notably, we demonstrate control over high-quality CCAs comprising up to 100 resonators with a density of 5 resonators per 100 $\mu$m, highlighting our capability to finely engineer the environment bandwidth.

To demonstrate a multiband spectrum, we consider configurations with $M$ up to 5 cavities per unit cell (Fig. 2d,e). We denote with $J_i$,

where $i = 1...M - 1$, the coupling between cavities within a unit cell (intracell coupling) and with $J_{M+1}$ the coupling between unit cells (intercell coupling). As we increase the number of elements in the unit cell, additional bands appear in the array spectrum[45]. As such, bandgaps are expected to emerge in the midst of the CCA's spectrum, with up to $M$ passbands. In the dimer case ($M = 2$), each resonator presents the same two coupling capacitances, in an alternating fashion, which automatically satisfies the resonant condition. However, for $M > 2$, the cavities do not necessarily have the same total capacitance and hence maintaining a constant resonant frequency for all cavities requires fine tuning of the inductors to compensate for this effect (see Supplementary file).

In all measurements in Fig. 2, the influence of higher-order coupling terms, $J'$, are present and manifest in two primary aspects: first, the mode distribution in the passband is asymmetric with respect to $\omega_r$, resulting in higher mode density at lower frequencies (see Supplementary file). Second, the mode coupling to the ports, $\kappa_{\mathrm{ext}}$, for the low frequency eigenmodes is lower than for their higher frequency counterpart (see Supplementary file). By fitting the CCA spectra (see Methods), we estimate $J' \approx 10\%\bar{J}$ for the rectangular designs (dominated by direct stray capacitive coupling), while for the hexagonal one, $J' \approx 20\%\bar{J}$ (due to high $C_{i,i+1}/C_{\Sigma,i}$ ratio), where $\bar{J}$ is the mean of the nearest neighbor couplings in the CCA (see Table in Fig. 2c). Furthermore, the asymmetry observed in the size of the bandgaps (Fig. 2e) can be attributed to systematic design imperfection (see Supplementary file).

## Engineering localized modes

Using our architecture, we demonstrated excellent control over the bulk spectrum of the CCA. Relying on topology, the creation of low-dimensional bound modes can prove useful for coupling to quantum emitters[46–49]. Remaining in a 1D chain geometry, we turn to engineer CCAs in the topologically non-trivial SSH configuration (Fig. 3)[36,37]. The SSH chain can be in a chiral-symmetry-protected topological phase, where its bands exhibit a quantized bulk polarization, with associated mid-gap 0D edge states. The model has been extensively studied in photonic CCAs[50], cold-atoms[51,52], polaritonics[53], and optomechanical arrays[54], and used to engineer directional topological waveguide QED[20,55].

Our microwave photonic analogue of the SSH model comprises a dimerized ($M = 2$) chain, with the unit-cell made of $A$ and $B$ sites (Fig. 3a). The intra- ($J_1$) and inter-cell ($J_2$) hopping are alternating, leading to a two bulk-band spectrum. However, a gap closing occurs when $J_1 = J_2$ (Fig. 3b). The gap closing marks a topological phase transition between the topologically-trivial ($J_2 < J_1$, cf. Fig. 2d) and the topologically nontrivial cases ($J_1 < J_2$, cf. Fig. 3). A quantized jump in the bulk polarization of the chain distinguishes between the two cases, where in the latter it implies the appearance of two degenerate mid-gap edge states.

The bulk Hamiltonian of our SSH CCA, in quasimomentum space reads[56]:

$$\hat{H}(k) = (\omega_0 + 2J' \cos(kd))\, \tau_0 \\ + (J_1 + J_2 \cos(kd))\, \tau_x + J_2 \sin(kd)\, \tau_y + \xi\, \tau_z \tag{3}$$

where $k$ is the reciprocal wavevector, $d$ is the lattice constant (distance between the unit cells), and $\tau_0$, $\tau_x$, $\tau_y$ and $\tau_z$ represent the Pauli matrices. While periodic boundary conditions were applied to derive the Hamiltonian in Eq. (3), to observe SSH edge states, the CCA must have open boundary conditions. Furthermore, the mid-gap edge states are well defined only if chiral symmetry is realized, i.e., when $\hat{\Gamma}\hat{H}\hat{\Gamma}^{\dagger} = -\hat{H}$ holds, where $\hat{\Gamma}$ is the chiral symmetry operator. Specifically, for the SSH model, $\hat{\Gamma}$ is defined using the sublattice projectors associated to the $A$ and $B$ sites, $\hat{P}_A$ and $\hat{P}_B$, as $\hat{\Gamma} = \hat{P}_A - \hat{P}_B$[37,57]. Hence, the appearance of next nearest-neighbor hopping $J'$, and disorder terms $\xi$ due to fabrication imperfections lead to deviation from the standard SSH model[56] (see Supplementary file). The latter breaks the chiral symmetry and will move the topological edge states away from the middle of the gap. Disorder in first neighbor hopping, $J_i$, does not break the chiral symmetry and will bear a lesser impact on the edge states[37] (see Supplementary file).

In a finite-size CCA (Fig. 3a), the tails of the mid-gap edge states overlap in the bulk, resulting into a finite hybridization that gives rise to a frequency splitting according to

$$\Delta_{\text{Topo}}^{N}/2 \propto e^{-(N-1)/\zeta}, \tag{4}$$

where $\zeta$ is the edge states' spatial localization

$$\zeta = \frac{1}{\log J_2/J_1}. \tag{5}$$

The degree of hybridization between the SSH edge states depends on the chain's size, $N$, and the coupling ratio, $J_2/J_1$. Correspondingly, the hybridized SSH edge states form symmetric and antisymmetric superpositions between the left and right edge states, see Fig. 3c,e. In the remainder of the manuscript, we will refer to the *hybridized SSH edge states* as *SSH edge modes*.

We experimentally investigate two distinct configurations: a weakly-localized case with $J_2/J_1 = 1.22$ (Figs. 3c,d) and a strongly-localized case with $J_2/J_1 = 1.57$ (Figs. 3e,f). These configurations present CCAs with $N = 16$, 32, and 64 resonators. Measurements of CCA transmission, $S_{21}$, reveal a significant reduction in the amplitude of the SSH edge modes as the size of the CCA increases. This reduction can be attributed to the decreasing overlap of the localized edge states in the bulk region, resulting in reduced coupling and, therefore, reduced transmission between the two microwave ports (see Supplementary file). This trend is also visible in the behavior of the phase shift of the SSH edge modes measured in reflection ($S_{11}$, $S_{22}$)[58] (right panel of Figs. 3d,f). As the modes' hybridization reduces, the phase shift of the SSH edge modes becomes more prominent, indicating a stronger coupling to the microwave ports due to localization at the boundary.

As for the topologically trivial CCAs (Sec. II B), next nearest neighbor coupling have an influence on the SSH-CCAs spectra. We expect that the two edge modes for $J' \neq 0$ do not exhibit anymore perfect localization on a single sub-lattice of the unit cell (see Supplementary file). Instead, some photonic population extends into the neighboring sub-cell, thereby breaking chiral symmetry, even in the absence of a $\tau_z$ term in the Hamiltonian in Eq. (3). For $J'$ smaller than the bandgap, the SSH edge modes retain partial protection[59,60]. $\Delta_{\text{Topo}}^{N}$ exponentially decreases as a function of $N$, even in the presence of non-zero $J'$.

However, as shown in the right panels of Fig. 3d,f, the measured splitting between the SSH edge modes does not always decrease as a function of $N$, contrarily to what is predicted by Eq. (4). In particular, for the specific realization of the $J_2/J_1 = 1.57$ case (Fig. 3f), the splitting $\Delta_{\text{Topo}}$, even appears to increase with $N$. As discussed in the next section, this effect is attributed to disorder.

## Disorder

We have demonstrated the scalability and versatility of the architecture, exhibiting a high degree of control and suggesting minimal resonant frequency scattering between the cavities. However, some amount of disorder remains evident in the spectra of the CCAs. In the bulk, disorder manifests in deviations from a smooth envelope profile of the mode's transmission and in displacement of mode frequencies from their expected dispersion relation (Figs. 2 and 3). Quantifying disorder from the bulk modes is challenging due to the system complexity. On the other hand, SSH edge modes, despite being localized at the edges of the CCA, are readily probed. Crucially, due to the bulk-edge correspondence[37] they are also sensitive to chiral symmetry-breaking disorder in the bulk of the CCA. Hence, they can serve as a reliable indirect probe to quantitatively assess the extent of disorder in the bulk, solely by analyzing the behavior of the two SSH edge modes.

To investigate the impact of disorder on the SSH edge modes, we fabricate and characterize additional SSH devices (see Supplementary file) with a coupling ratio of $J_2/J_1 = 1.22$, as illustrated in Fig. 3d. In Fig. 4a, we present a dataset comprising $\Delta_{\text{Topo}}^{N}$ values extracted from the measured CCAs as a function of chain length $N$. Notably, while $\Delta_{\text{Topo}}^{N}$ exhibits the expected exponential decay with respect to $N$, it does not asymptotically approach zero, demonstrating significant deviations from the theoretical prediction (black line in Fig. 4a).

To rigorously account for this observation, we conduct numerical simulations that introduce Gaussian noise, denoted as $\sigma_L$, applied to the inductance values of all resonators within the chain (red line in Fig. 4a). The choice to introduce scattering in $L_g$ as the main noise source is motivated by the fact that the inductors have the most critical dimension in the resonator design, rendering them more susceptible to scattering during the fabrication process. The $\sigma_L$ noise applied to the inductors induces both $\tau_z$ and $\tau_x$ type of disorder in Eq. (3), impacting respectively the resonant frequency and the coupling of the resonators in the CCA. Our analysis primarily focuses on the principal effect of $\sigma_L$ scattering namely $\tau_z$-type disorder (breaking chiral symmetry), which we refer to as $\sigma_{L \to z}$. These simulations accurately describe the behavior of $\Delta_{\text{Topo}}$ observed in Fig. 3d, f. In the inset of Fig. 4a, we present the simulated Probability Density Function (PDF) for $\Delta_{\text{Topo}}^{64}$ as a function of $\sigma_{L \to z}$ (see Methods). For $\sigma_L = 0$, $\Delta_{\text{Topo}}^{64} \sim 0$, with $J_2/J_1 = 1.22$. As the disorder increases, the probability of observing $\Delta_{\text{Topo}}^{64}$ values higher than $\sim 0$ also increases, along with the standard deviation.

In the measurements shown in Fig. 4a, a notable deviation is observed for $\Delta_{\text{Topo}}^{22}$ compared with the general trend. To gain further insight into the source of this deviation, we perform time-resolved measurements of the SSH edge modes amplitude, using the pulse sequence illustrated in Fig. 4b. This sequence involves sending a Gaussian pulse at a frequency between the two SSH edge modes, leading to beating oscillations in time between the modes' population if the two modes are hybridized, with a frequency of the beating corresponding to the coupling rate $\Delta_{\text{Topo}}^{N}/2$ (see Supplementary file). The

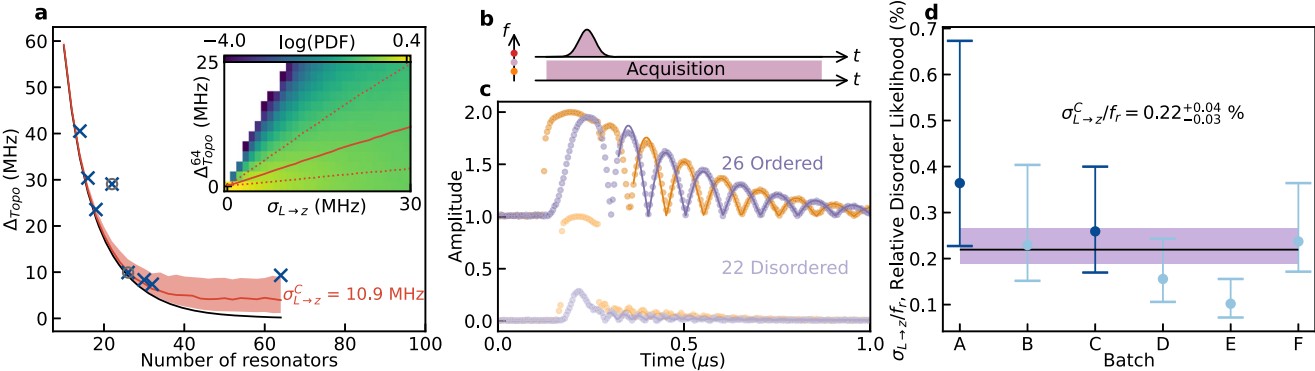

**Fig. 4 | Using topology to study disorder. a** Study of the frequency splitting, $\Delta_{\text{Topo}}^{N}$, between the hybridized SSH edge modes for CCAs with different $N$ ($J_2/J_1 = 1.22$). The cross and the circles show $\Delta_{\text{Topo}}^{N}$ extracted for CCAs with $N = 14, 16, 18, 22, 26, 30, 32$ and $64$, respectively from spectroscopy and time resolved measurements. The continuous black line represents the expected evolution of $\Delta_{\text{Topo}}^{N}$ *vs* $N$ for the disorderless case according to the eigenmodes of the CCAs Hamiltonian (see Methods). The continuous red line represents the median of the most likely evolution (see panel **d**) of $\Delta_{\text{Topo}}^{N}$ *vs* $N$ in the presence of disorder with standard deviation $\sigma_{L \to z}^{C} = 10.3$ MHz applied to $\omega_r$ of the cavities in the CCAs. The shaded region represents the $1\sigma$ uncertainty on the estimation of $\Delta_{\text{Topo}}^{N}$. The inset shows the simulated logarithm of the Probability Density Function (PDF) of $\Delta_{\text{Topo}}^{64}$ as a function of $\sigma_{L \to z}$. The red line shows the median for each disorder realization. The red dotted lines show the $1\sigma$ standard deviation. **b** Pulse sequence used for the

time-resolved measurement. A gaussian pulse is sent at a frequency between the hybridized SSH edge modes from one of the edge of the CCA, while the signal is acquired on both side of the CCA. The orange (red) dot on the *y*-frequency axis on the left side, highlights the frequency of the antisymmetric (symmetric) hybridized SSH edge modes. **c** Time traces of the transmitted $|S_{21}|$ and reflected $|S_{11}|$ signals of the CCAs with $N = 26$ and $22$ in panel (**a**). Each data points is averaged 20000 times. The continuous lines highlight the fit done with an exponentially decaying cosine (see Supplementary file). **d** Disorder likelihood extracted for different batches of devices. The dots highlight the maximum likelihood extracted for each batch. The error bars show the respective FWHM. The black line shows the combined likelihood extracted among the different batches. The shadowed purple area shows its FWHM. The color code is according to Fig. 3d,f. Batch $A$ ($C$), contains the CCAs with $N = 16, 32$ and $64$ ($14, 18, 22, 26, 30$) of panel (**a**).

results of this measurement are presented in Fig. 4c for the cases corresponding to the two circled data points in Fig. 4a: $N = 22$ (the outlier point in Fig. 4a) and $N = 26$ (a representative of devices following the trend in Fig. 4a). For the $N = 26$ SSH-CCA, we distinctly observe beatings between the two SSH edge modes at a frequency of approximately 4.97 MHz, aligning with the value of $\Delta_{\text{Topo}}^{26} = 4.96$ MHz extracted from spectroscopy measurements (Fig. 4a). In contrast, for the $N = 22$ CCA, we observe a significantly reduced visibility of the beating pattern, indicating weak coupling to the edge microwave ports and, therefore, suggesting that the modes are localized not at the edge but more in the bulk of the CCA. This could be due to two effects: a strong disorder at the edge lifts the resonant frequency of a resonator or a strong impurity in the bulk (strongly detuned resonator) effectively divides the chain and quenches the transmission. We utilize these time-resolved measurements to identify devices with strong local disorder that no longer conform to our Gaussian disorder model of the SSH.

To evaluate the overall disorder introduced during the fabrication process, we conduct a comprehensive statistical analysis involving 26 CCAs in the SSH configuration. These 26 devices, manufactured during different fabrication runs, have been designed in the two SSH configurations depicted in Fig. 3 (see Supplementary file). To quantify the disorder, we extract $\Delta_{\text{Topo}}^{N}$ from spectroscopy measurements for all the tested devices and generate their associated PDFs as functions of $\sigma_L$ and $N$ for each sample batch $S$. For each batch, we compute the likelihood of the inferred frequency fluctuation with respect to the model as a function of $\sigma_L$, which we refer to as likelihood function, defined as follows:

$$\mathcal{L}_S(\sigma_L) = \prod_i \text{PDF}_S(N_i, \Delta_{\text{Topo},i}^{N_i}; \sigma_L), \qquad (6)$$

where $N_i$ and $\Delta_{\text{Topo},i}^{N_i}$ represent respectively the number of resonators in the CCA and the SSH mode frequency splitting of the $i^{\text{th}}$ CCA. We report the extracted maximum likelihoods and their full width at half maximum (FWHM) in Fig. 4d. To obtain an estimate of the disorder across all the tested batches, we computed the combined likelihood, represented

by the black line in the same figure (see Methods). Our analysis yield an extracted relative disorder value of $\sigma_{L \to z}/f_r = 0.22^{+0.04}_{-0.03}\%$, equivalent to an absolute disorder value of $\sigma_{L \to z}^{C} = 10.97^{+2.28}_{-1.59}$ MHz for $\omega_r/2\pi \sim 5$ GHz. This represents a minimal frequency scattering, especially considering the high compactness of the implemented CCAs, and it is comparable to what is achieved with lattices of CPW resonators (but with $10^{2-3}$ larger footprint)[16] and state-of-the-art frequency scattering control of advanced MKIDs detector arrays[61,62].

## Discussion

We have presented a platform based on coupled cavity arrays (CCAs) utilizing high kinetic inductance NbN thin films, which serve as compact multipurpose high-impedance metamaterials in the microwave domain. The compactness of each cavity allows for the integration of 1D CCAs with up to 100 resonators within a few millimeters of sample space. The remarkable versatility of our CCA platform has been demonstrated through the creation of CCAs with bandwidths ranging from a few 100 MHz up to 4.5 GHz and the engineering of multiple bandgaps. Importantly, all fabricated devices exhibited mode dispersion in excellent agreement with our exact models. Furthermore, using the SSH chain's in-gap modes, we extracted a small resonator frequency scattering of $\sigma_{L \to z}^{C}/f_r = 0.22^{+0.04}_{-0.03}\%$.

Our findings pave the way for advancing technological applications and fundamental investigations using multimode light-matter systems. This platform will allow for a straightforward extension to very large-scale 1D and 2D multimode systems with up to $10^4$ cavities on a single $5 \times 5\text{mm}^2$ chip. The versatility of our platform in controlling mode densities presents exciting prospects for exploring devices where emitters are coupled to high-impedance multimode environments[7]. This provides the means to study the ultrastrong coupling limit in both the passband of the CCA[63] and atom-photon bound states close to the band edges[25,64]. This will allow to investigate quantum-impurity models like spin-boson[65] or Frenkel-Holstein[17,18] type Hamiltonians. Additionally, the compact nature of the resonators facilitates coupling to superconducting qubits at multiple points, potentially with non-trivial phase delays.

This makes our architecture a natural platform for studying giant-atom physics[66] and giant-atom photon-bound states in structured environments[67,68].

On the other hand, photon lattices also offer promising avenues for future experiments aimed at investigating quantum phase transitions[1,69]. By effectively reducing random disorder, cavity arrays can be fabricated with controlled levels of disorder, potentially enabling the study of many-body localization effects[44]. While our current work remains non-interacting, the incorporation of interactions is feasible through the inherent nonlinearity present in high kinetic inductance materials, resulting in both $\chi^2$ and $\chi^3$ nonlinearities[70], or by integrating qubits into each resonator[69]. We aim to leverage nonlinearities in CCAs to investigate driven-dissipative phase transition[69,71,72]. Moreover, these lattices facilitate the creation of unique devices capable of hosting photons in curved spaces[19], gapped flat band[73,74], and enable alternative forms of qubit-qubit interaction[20,75]. The high-impedance, ultracompact nature of our platform enables the realization of dense, multimode environments[7,8,41], providing a pathway to study quantum many-body phenomena with a high degree of control. Its scalability, without sacrificing quality or increasing disorder, paves the way for future explorations of light-matter interactions in unconventional regimes, such as multimode ultra-strong coupling[9,43,44] and complex photonic states for analog quantum simulation[17,18,24]. Additionally, our *topological disorder-meter* provides a versatile tool for characterizing disorder in a wide range of quantum or classical systems, and is, in principle, applicable to any symmetry-protected topological phase, such as Kitaev chains[76,77] or 2D graphene-like lattices[78]. These advancements position our platform as a flexible foundation for advancing quantum technologies[2,3] and specifically analog quantum simulation[17,18,24].

## Methods

### Fabrication

The fabrication recipe is detailed in[35]. We fabricate planar coupled cavity arrays (CCAs) based on lumped LC resonators by etching 13 nm-thick NbN film, with typical sheet kinetic inductance $L_{k,\square}$ of 100 pH/$\square$. The fabrication process commences with a 2-minute immersion in a 40% HF bath to eliminate the native oxide layer and potential surface impurities from a 100 mm silicon wafer, which is of high resistivity ($\geq 10 k\Omega$cm) and has a $\langle 100 \rangle$ orientation. Using a Kenosistec RF sputtering system at room temperature, NbN films are bias sputtered following the method described in refs. 35,79 with Ar/N$_2$ flows of 80/7 sccm respectively and a deposition pressure of $5\,\mu$bar. Optical lift-off technique is employed to deposit Ti/Pt alignment markers, followed by a dehydration step at 150° C for 5 minutes. 80 nm-thick CSAR positive e-beam resist is then spun on the wafer, which is subsequently baked at 150° C for 5 minutes. Employing electron beam lithography (Raith EBPG5000+ at 100 keV), the resist is patterned to form the desired devices. This is achieved by developing the resist in amyl acetate for 1 minute, followed by rinsing in a 9:1 MiBK:IPA solution. In order to transfer the pattern onto the NbN, a reactive ion etching process is employed using a CF$_4$/Ar mixture. The etching is carried out with a power of 15 W, using a stepped approach consisting of 10 steps, each lasting for 1 min. These etching steps are alternated with 1-minute purges using Ar gas. This stepped etching technique has proven advantageous as it reduces the damage caused to the CSAR resist due the etching process, thereby facilitating its subsequent stripping without the need for plasma oxygen, which may damage the underlying superconductor. The resist is then stripped using Micoposit remover 1165 heated to 70° C. Finally, the wafer is coated with a 1.5 $\mu$m AZ ECI 3007 positive photolithography resist to protect the devices before being diced.

## Model

In this section, we derive the Hamiltonian of the CCAs using standard circuit quantization[42]. We consider a chain composed of $N$ capacitively coupled LC resonators as depicted in Fig. 1. Each $i^{\text{th}}$ resonator possesses an inductance $L_{ig}$ connected to ground and capacitance $C_{ig}$ to ground. Resonators $i$ and $j$ are mutually coupled via the coupling capacitance $C_{i,j}$ between the two resonators. The potential energy in the inductors can be expressed as

$$E_L = \frac{1}{2}\sum_{n=1}^{N}\frac{\phi_n^2}{L_{ng}}, \tag{7}$$

where $\phi_n$ denotes the flux at node $n$. The total kinetic energy stored in the chain's capacitors is given by

$$E_C = \frac{1}{2}\left[\sum_{n=1}^{N}C_{ng}\dot{\phi}_n^2 + \sum_{i,j}C_{i,j}(\dot{\phi}_i - \dot{\phi}_j)^2\right], \tag{8}$$

where $\dot{\phi}_n$ represents the electric potential at node $n$. We neglect mutual inductance-induced coupling due to the high impedance of the resonators[38]. We can now write the Lagrangian, $\mathbb{L}$, of the circuit as

$$\mathbb{L} = E_C - E_L \tag{9}$$

$$= \frac{1}{2}\sum_{n=1}^{N}\left[C_{ng}\dot{\phi}_n^2 - \frac{\phi_n^2}{L_{ng}}\right] + \frac{1}{2}\sum_{i,j}C_{i,j}\left(\dot{\phi}_i^2 - \dot{\phi}_j^2\right). \tag{10}$$

It can be written in a matrix form as

$$\mathbb{L} = \frac{1}{2}\dot{\boldsymbol{\phi}}^T[C]\dot{\boldsymbol{\phi}} - \frac{1}{2}\boldsymbol{\phi}^T\left[L^{-1}\right]\boldsymbol{\phi}, \tag{11}$$

with the vectors $\dot{\boldsymbol{\phi}}_n^T = (\dot{\phi}_1, \dot{\phi}_2, \ldots, \dot{\phi}_N)$ and $\boldsymbol{\phi}_n^T = (\phi_1, \phi_2, \ldots, \phi_N)$. The capacitance matrix is defined as

$$[C]_{ij} = \begin{cases} C_{\Sigma i}, & \text{if } i=j, \\ -C_{i,j}, & \text{if } i \neq j \text{ and } |i-j| \leq 3, \\ 0, & \text{if } i \neq j \text{ and } |i-j| > 3, \end{cases} \tag{12}$$

where we only consider mutual capacitances where $|i-j| \leq 3$. $C_{\Sigma i}$ is the total capacitance of the $i^{\text{th}}$ cavity defined as

$$C_{\Sigma i} = C_{ig} + \sum_{n \neq i}^{N}C_{i,n}. \tag{13}$$

The inverse inductance matrix is defined as

$$[L^{-1}]_{ij} = \begin{cases} 1/L_{ig}, & \text{if } i=j, \\ 0, & \text{if } i \neq j. \end{cases} \tag{14}$$

We now introduce the node charge variable canonically conjugated to the node flux $\phi_n$

$$\boldsymbol{Q} = \frac{\partial \mathbb{L}}{\partial \dot{\boldsymbol{\phi}}}, \tag{15}$$

with $\boldsymbol{Q}^T = (Q_1, Q_2, \ldots, Q_N)$.

For the given system, the charge variables are $\boldsymbol{Q} = [C]\dot{\boldsymbol{\phi}}$. Making use of the matrix formalism, the CCA Hamiltonian $H$ then reads

$$H = \frac{1}{2}\boldsymbol{Q}^T[C^{-1}]\boldsymbol{Q} + \frac{1}{2}\boldsymbol{\phi}^T[L^{-1}]\boldsymbol{\phi}. \tag{16}$$

The real space Hamiltonian can be found to be[42]

$$\boldsymbol{H_n}/\hbar = \sqrt{[C^{-1}][L^{-1}]}, \tag{17}$$

and have the following matrix form

$$\frac{\boldsymbol{H_n}}{\hbar} = \begin{pmatrix} \omega_1 & J_{1,2} & J_{1,3} & \cdots & \cdots & J_{1,N} \\ J_{2,1} & \omega_2 & J_{2,3} & \ddots & \ddots & \vdots \\ J_{3,1} & J_{3,2} & \omega_3 & \ddots & \ddots & \vdots \\ \vdots & \ddots & \ddots & \ddots & \ddots & \vdots \\ \vdots & \ddots & \ddots & \ddots & \omega_{N-1} & J_{N-1,N} \\ J_{N,1} & \cdots & \cdots & \cdots & J_{N,N-1} & \omega_N \end{pmatrix}. \tag{18}$$

For small $C_{i,i+1}/C_{\Sigma,i}$ ratio, and without stray next nearest neighbor capacitances, the coupling is

$$J_{i,j} = \frac{\sqrt{\omega_i \omega_j}}{2} \frac{C_{i,j}}{\sqrt{C_{\Sigma,i} C_{\Sigma,j}}}. \tag{19}$$

However, non-negligible additional contributions arise from both parasitic capacitances $C_{i,l \neq j}$ and large $C_{i,i+1}/C_{\Sigma,i}$ ratio, modifying the Hamiltonian (see Supplementary file). In that case, one cannot resort to simple analytical formulas and a full numerical diagonalization is required. Eigenvalues of Eq. (18) represent the frequencies of the modes $\omega_k$ and eigenvectors represent the spatial localization of the modes. We utilize this model to fit the CCA modes' frequencies, as detailed in the following section.

## Extraction of parameters

The estimation of the parameters is performed by extracting the modes' frequencies from the device's spectrum and fitting them to the eigenvalues of Eq. (18). The fitting process involves 5 to 8 fitting parameters, depending on the specific design, i.e. if the CCA is dimerized, trimerized, etc …. Initially, we make the assumption that each fitted CCA is disorder-free and uniform. The influence of disorder is studied in the following section. The fitting parameters include:

- $C_g$, the capacitance to ground. For a single resonator ($M = 1$) or a dimer ($M = 2$), the capacitive environment is automatically identical for each resonator and we use a single value for $C_g$. However, for $M > 2$, the capacitive environment of each cavity is not identical (see Supplementary file) and $C_g$ becomes a list, $\overrightarrow{C_g}$, comprising the different $C_{ig}$ within a unit cell.
- $L_g$, the inductance to ground. For $M \leq 2$, since the capacitive environment is similar for each cavity, having a constant inductance ensures a constant frequency for each cavity in the unit cell and we use a single value for $L_g$. For $M > 2$, in order to keep the resonant frequency constant, $L_g$ is adjusted for each cavity (see Supplementary file). In this case, it becomes a list $\overrightarrow{L_g}$.
- $\overrightarrow{C_c}$, the coupling capacitances, which form a list increasing with the size of the unit cell, i.e. for $M = 2$, $C_c = (C_1, C_2)$, $C_1$ and $C_2$ being the intra- and inter-cell capacitances.
- $C_{i,i+2}/\overline{C}$, the ratio of second neighbor coupling capacitances over $\overline{C}$, the mean value of $\overrightarrow{C_c}$.
- $C_{i,i+3}/\overline{C}$, the ratio of third neighbor coupling capacitances over $\overline{C}$.

For the fits to converge, we must either fix $L_g$ or $C_g$ as they both contribute comparably to the resonant frequency of the cavities and the coupling between cavities. To this end, we choose to fix $L_g$. We determine the value of $L_g$ through finite-element microwave simulations, where we estimate the sheet kinetic inductance of the film using Sonnet simulation software. This process involves three steps:

1. We initially fit the modes of the measured spectrum with the eigenvalues of Eq. (18), fixing $L_g$ with an initial guess. This provides a precise estimation of the resonant frequency of the cavities but not of the other parameters.

2. We then conduct a simulation for a single cavity in Sonnet, while sweeping the kinetic inductance, $L_{k,\square}$. The simulated cavity has a capacitive environment equivalent to that of the cavities in the fitted CCA, ensuring accurate estimation of the resonant frequency. Subsequently, we extract and fit the resonant frequency of the cavity as a function of the kinetic inductance using the following function for the frequency:

$$f = \frac{1}{2\pi \sqrt{(L_{k,\square} \frac{l_{\text{ind}}}{w_{\text{ind}}} + L_{\text{geo}}) C_\Sigma}}, \tag{20}$$

where $C_\Sigma$ represents the total capacitance of the resonator and $L_{\text{geo}}$ is the geometric inductance. The extracted $L_{\text{geo}}$ is usually 2 orders of magnitude smaller than $L_k$. The parameters $l_{\text{ind}}$ and $w_{\text{ind}}$ are fixed design parameters, corresponding to the length and width of the inductor, respectively. This procedure allows us to fit $L_g = L_{k,\square} \frac{l_{\text{ind}}}{w_{\text{ind}}}$. To extract $L_{k,\square}$ it is necessary to properly estimate the dimension of the inductor via an SEM of the device.

3. Using the obtained value of $L_{k,\square}$, we calculate the inductance to ground $L_g$ using Eq. (1) and then refit the measured spectrum with the correct $L_g$ as a fixed parameter. This enables us to determine the values of $C_g$, $\overrightarrow{C_c}$, $C_{i,i+2}/\overline{C}$, and $C_{i,i+3}/\overline{C}$. It is important to note that the estimation of parameters using this method is affected by disorder in the CCA, which introduces a small systematic error (see Supplementary file).

## Disorder estimation

In this section, we outline the procedure for extracting the level of disorder from the frequency splitting of the hybridized SSH edge modes, $\Delta_{\text{Topo}}^N$.

The study is performed on six different batches realized in different fabrication runs. Each batch comprises three to six devices with different number of resonators. Two batches, A and C, are designed to have a coupling ratio $J_2/J_1 \approx 1.22$, while the others batches are designed to have $J_2/J_1 \approx 1.57$. We observed changes in $L_{k,\square}$ due to potential fluctuations in the film deposition process (thickness or composition), resulting in up to 15% change of the CCA resonant frequency, $\omega_r/2\pi$. This variation of kinetic inductance has been reduced to ~ 3% with improved monitoring and deposition adjustments. Those change have a minimal effect on $J_2/J_1$.

We initially employ the fitting routine presented in the previous Methods section to extract the mean capacitances and inductances specific to each batch, assuming a disorder-free scenario.

Then we utilize these parameters in the Hamiltonian Eq. (18) where we introduce Gaussian noise with a standard deviation $\sigma_L$ applied to the inductances:

$$\boldsymbol{H_n}/\hbar = \sqrt{[C^{-1}][L^{-1}(\sigma_L)]}. \tag{21}$$

Due to the high kinetic inductance of the films and the small size of the inductors, the inductances are sensitive to fabrication imperfection. Hence, the Gaussian noise, $\sigma_L$, is applied to the inductances of the cavities. For each batch, number of resonators $N$ and disorder value $\sigma_L$, we generate 30,000 realizations of the Hamiltonian Eq. (21). We then diagonalize each of the Hamiltonians and extract $\Delta_{\text{Topo}}^N$. Using these simulations, we construct for each batch a three-dimensional probability density function (PDF) that depends on the number of resonators, the SSH-modes splitting and the level of disorder (Fig. S1a of Supplementary file).

Several insights can be derived from these PDF. First, as expected, the splitting between the SSH edge modes decreases with an

increasing number of resonators. Secondly, the PDF exhibit an asymmetry which tends to increase the splitting as a function of disorder. This asymmetry arises from the fact that disorder can only increase the splitting between the SSH edge modes. However, it is noteworthy that for short CCAs, such as the case with 16 resonators shown in Fig. S1a of Supplementary file, the splitting can also decrease with increasing disorder. This occurs when the SSH edge modes enter the bulk for sufficiently large disorder values. Thirdly, when $\Delta_{\text{Topo}}^N$ approaches zero, deviation from its expected value become significantly more prominent.

We proceed to compute the likelihood for each batch $S$, using an interpolated PDF, defined as follows:

$$\mathcal{L}_S(\sigma_L) = \prod_i \text{PDF}_S(N_i, \Delta_{\text{Topo},i}^{N_i}; \sigma_L) \qquad (22)$$

where $N_i$ and $\Delta_{\text{Topo},i}^{N_i}$ represent the number of resonators and the SSH edge modes frequency splittings of data point (device) $i$, respectively. $\text{PDF}_S$ is the Probability density function used for batch $S$. The likelihood functions (Fig. S1b of Supplementary file) are then normalized by their area from which we extract the full width at half maximum of the different likelihoods.

To obtain an overall assessment of disorder across all devices of different batches, we employ the method of combined likelihood, defined as follows:

$$\mathcal{L}^C(\sigma_L) = \prod_S \mathcal{L}_S(\sigma_L), \qquad (23)$$

where $S$ represents the label of the batch. This function characterizes the typical disorder among all fabricated devices in the topological configuration and is illustrated in Fig. S1c of Supplementary file. It is important to note that this method also presents some limitations, as it is sensitive to the accuracy of the estimation of the CCA parameters. Errors in the estimation of the coupling capacitances, for example, can lead to significant changes in the decay of the SSH edge modes. One way to mitigate this sensitivity is to operate in a regime where the hybridization between the modes is weak, reducing the impact of parameter misestimation.

## Data availability
The datasets generated during the current study are available in Zenodo with the identifier https://doi.org/10.5281/zenodo.14640126.

## Code availability
The codes used to produce the plots and perform the disorder analysis are available in Zenodo with the identifier https://doi.org/10.5281/zenodo.14640126.

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

## Acknowledgements

The authors thank Jan Košata and Andrea Bancora for stimulating discussions. The authors thank the EPFL physics mechanical workshop for help in setting up the lab. P.S. acknowledges support from the Swiss

National Science Foundation (SNSF) through the grants Ref. No. 200021_200418, Ref. No. 206021_205335, and project UeM019-16 – 215928, and from the Swiss State Secretariat for Education, Research and Innovation (SERI) under contract number 01042765 SEFRI MB22.00081. P.S also acknowledge support from the NCCR Spin Qubit in Silicon (NCCR-SPIN) Grant No. 51NF40-180604. S.F. acknowledges the support of SNF Spark project 221051. M.S. acknowledges support from the EPFL Center for Quantum Science and Engineering post-doctoral fellowship. O.Z. acknowledges funding from the Deutsche Forschungsgemeinschaft (DFG) via project number 449653034 and through SFB1432, as well as the Swiss National Science Foundation (SNSF) through the Sinergia Grant No. CRSII5_206008/1. All devices were fabricated in the Center of Micro-NanoTechnology (CMi) at EPFL.

## Author contributions

V.J., S.F., V.J.W. and P.S. designed the experiment. V.J., S.F., F.O., F.D.P. and D.S. fabricated the devices. S.F. and F.O. developed and optimized the fabrication recipes. S.F. developed and optimized the NbN deposition. V.J. performed the measurements. V.J. analyzed the data with supervision from L.P. and inputs from G.B. V.J. numerically simulated the model with supervision from M.S. P.S. and O.Z. supervised the experimental and theoretical parts of the project. V.J., S.F., L.P., M.S., O.Z. and P.S. contributed to the writing of the paper.

## Competing interests

The authors declare no competing interests.
