## [Peer Review File · Nature Communications]

High kinetic inductance cavity arrays for compact band engineering and topology-based disorder meters

Corresponding Author: Mr Vincent Jouanny

Version 0:

Reviewer comments:

Reviewer #1

(Remarks to the Author)

The authors investigate the use of high kinetic inductance materials for the fabrication of coupled cavity arrays (CCAs). CCAs have been proposed as platforms for quantum simulation as cavity frequencies and couplings can be systematically varied. Early work with CCAs [refs. 15, 18] examined large 2D arrays, and while the research was motivated by interesting physics, the experimental results were rather ambiguous. The present manuscript by Jouanny et al. takes a needed step back and carefully examines 1-dimensional (1d) cavity arrays fabricated with NbN, a high kinetic inductance material.

The research team first characterizes CCAs with one cavity per unit cell. With the exception of some slot line modes at high frequencies, the cavity transmission spectrum is consistent with expectations. They then investigate CCAs with more than one cavity per unit cell. These measurements show the expected bands and bandgaps. Lastly, CCAs with staggered coupling rates are fabricated to investigate the SSH model. The expected topological modes (and their hybridization) are observed and characterized using a novel time-domain measurement. A major outcome of the manuscript is the characterization of the disorder, which is about 11 MHz, as compared with the cavity frequency ~ 5 GHz.

Obviously, a large collection of coupled harmonic oscillators is going to exhibit a complex normal mode spectrum. The present manuscript goes beyond refs. 15 and 18 in that it demonstrates a deep understanding of the measured normal mode spectrum. As such, I strongly encourage publication in Nature Communications after the authors address a few questions.

- The authors refer to a hexagonal geometry (Figs. 1 c,d), but upon examining the device architecture, it appears to just be a 1d CCA that is patterned in a zig-zag pattern. Specifically, it only consists of a 1d array of resonators, with each resonator coupled by an inter-digitated capacitor. It could of course be extended to a larger 2D hex array, but this is not demonstrated in the manuscript. As the results from the "hex" geometry are not central to the paper, I encourage the authors to more accurately describe the geometry in Figs. 1(c,d), or remove these panels. As is, the results are somewhat overstated.

- The authors refer to "ghost ports" near the end of section 1A. While the need to engineer parasitic capacitances to maintain uniform cavity frequencies is apparent, the manner in which this is achieved is unclear from the text and Fig. 1b. I therefore appreciate it if the authors can comment in more detail on how parasitics at the ends of the CCA are managed.

- Minor: a) Consider removing Figs. 1c,d and consolidating Fig. 1. b) There appears to be an inconsistency between the legend of Fig. 1c, where $J/2\pi = 164$ MHz is listed, and the caption, where $J/2\pi = 200$ MHz is listed. c) The formatting of the references is somewhat muddled, e.g. 2, 4, 5, 22, 30, ... The references should be cleaned up before resubmission.

Reviewer #2

(Remarks to the Author)

The manuscript by Jouanny et al (NCOMMS-24-22592-T) presents a new experimental circuit-QED architecture for implementing large-size photonic lattices described by tight-binding models (and potentially able to be coupled to artificial atoms). The setup consists of an array of capacitively coupled resonators made of high kinetic inductance NbN thin film, which compared to state-of-the-art devices allow for a significant size reduction, so that the authors can assemble up to 100

resonators within a few mm^2 chip. This is achieved by keeping the amount of disorder comparable with that of larger devices.

The authors in particular realise and test (mostly through transmission spectra):

- multi-band lattices (up to five bands) featuring multiple bandgaps;
- the SSH model in the topological non-trivial phase;
- edge modes occurring in the SSH model when this is in a non-trivial topological phase for a varying number of resonators in the array.

Edge modes, and especially their frequency splitting, are studied as a function of disorder and this is proposed as an interesting method to measure the disorder amount of the entire array relying on the bulk-edge correspondence principle.

I think the work is a high-quality contribution to the concerned literature, marking a milestone that has the potential to spur further advancements in the field, both of experimental and theoretical nature. I particularly appreciated the presentation, which is very clear and accurate (including the three figures in the main text). For these reasons, I support publication on Nat Comm only provided that the following points are addressed by the authors.

- 1) Given its importance in the present work, I believe it is useful for the reader to recall explicitly the definition of chiral symmetry and the definition of the chiral symmetry operator for the SSH model;
- 2) Among platforms beyond circuit QED for investigating atoms/emitters coupled to photonic lattices, the Introduction mentions Ref. [10] dealing with atoms in photonic crystals. It is probably fair mentioning as well cold atoms in optical lattices, see e.g. Krinner et al, Nature 559, 589 (2018);
- 3) Ref. [66] does not deal with "giant-atom photon bound states", while these are indeed featured in Ref. [67] but only for the simplest special case of a homogeneous (one-band) lattice. The general theory of these states for an arbitrary photonic lattice (which complies very well with the spirit and outlook of the present manuscript) was recently presented in L. Leonforte et al., arXiv:2402.10275.

Some very minor suggested corrections on the text follow.

- page 3: "arise from two origins"->"have a double origin";
- end of column 1, page 3, add "quality factor" before Q; shortly after "doesn't"->"does not";
- page 4, column 1: the sentence "The SSH chain is a ubiquitous chiral symmetry protected topological phase of matter that manifest in 1D systems" is a bit awkward (a chain is not a phase; also I do not see what ubiquitous means). I would just replace it with "The SSH chain can be in/can host a chiral-symmetry-protected topological phase";
- to comply better with conventional terminology, I would recommend using "edge modes" (or "edge states", even though "modes" is more correct), instead of "boundary modes". Somewhat similarly, I would recommend replacing "SSH states" with "SSH edge states" (or "modes") which is still a rather short name.

Reviewer #3

(Remarks to the Author)

In this paper, the authors characterize high-kinetic-inductance coupled cavity arrays using NbN thin films, achieving a low footprint, precise control over circuit parameters, and low-frequency disorder. They fabricate devices with up to 100 cavities in different lattice configurations to study devices with multiple passbands. They then quantify the level of disorder in cavity arrays using the hybridization between edge modes of the topologically non-trivial SSH chain. Overall, the paper is interesting and well-written, with suitable physical intuition provided for understanding these large multimode systems. The work supports the main claims of the text, and the results are communicated effectively.

In my estimation, the main advancements of this work are (1) improved parameter control and miniaturization of superconducting metamaterials and (2) a novel method of quantifying disorder in the bulk. These improvements, while important and useful, are highly technical. The Su-Schrieffer-Heeger lattice has been studied previously using superconducting resonators, and multiple previous works have demonstrated coupling quantum emitters to coupled-cavity-arrays, even investigating qubit-qubit interactions in this setting. Because this work is mainly focused on technical improvements and the linear physics of metamaterials, I believe a more specialized journal would be more suitable. Below, I have attached some comments and questions.

1. Can the authors comment on the origin of chip slot modes? Does this refer to slot-line waveguide modes (as discussed in <https://pubs.aip.org/aip/apl/article/104/5/052602/828862/Fabrication-and-characterization-of-aluminum>) or packaging modes? In the former case, does the grounding wire between resonators (in the rectangular design) fail to suppress these modes?
2. Can the authors comment on whether the large kinetic inductance fraction of the resonators results in a strong sensitivity in the resonator frequency to temperature due to the thermal population of quasi-particles? Does this contribute to the disorder?
3. When discussing adding multiple resonators/unit cells, the term "resonant condition" is used, which refers to realizing resonators with degenerate frequencies in the entire chain. I believe the wording can be made clearer here to get this point across – an initial read led me to think this was a more involved requirement.

4. Is there a typo in the second term of Eqn. (3)? This term is not Hermitian. I believe it should be something like $(J_1 + J_2 \cdot \cos(kd)) \cdot T_x + J_2 \cdot \sin(kd) \cdot T_y$. A related comment is that this Hamiltonian, strictly speaking, refers to the SSH chain when periodic boundary conditions are applied. In other words, in the absence of disorder and next-nearest-neighbor hopping, there is still no zero-mode (edge mode) in the spectrum of this Hamiltonian. I think some clarifying statements on this would be helpful for readers.

5. Based on Eqn. (4), we expect the splitting between the edge modes to reduce with larger N . This is true for the weakly localized chain ($J_2/J_1 = 1.22$) but does not appear to be true for the strongly localized chain ($J_2/J_1 = 1.57$) in Fig. 3(d, f). Can the authors comment on what is happening?

6. We also expect that the edge-mode splitting should be reduced when the localization increases, based on Eqn. (5). However, in Fig. 3(d, f), this does not appear to be the case in the $N = 32$ and $N = 64$ cases. Can the authors comment on what is happening?

7. Can the authors comment on why the edge mode splitting can be fully explained by disorder in the frequency (T_z)? Can they rule out the possibility that other effects (beyond the SSH model) might play a role? For example, could coupling to a slot-line mode or packaging mode result in any hybridization? The role of T_x disorder is quite minimal, as shown in Fig. 14. What about disorder in J' (next-nearest-neighbor coupling, appearing in T_0)?

Reviewer #4

(Remarks to the Author)

Version 1:

Reviewer comments:

Reviewer #1

(Remarks to the Author)

The authors have modified the manuscript in response to suggestions from referees 1 and 2. I tend to disagree with referee 3, as the manuscript is one of the strongest "cavity lattice" papers published to date. As such, I recommend publication as is in Nature Communications.

Reviewer #2

(Remarks to the Author)

The authors satisfactorily addressed all of my concerns. I therefore recommend the manuscript for publication.

Reviewer #3

(Remarks to the Author)

I appreciate the authors' efforts to address my comments. I am satisfied with the answers to my technical questions.

However, my assessment of the scope impact of this manuscript remains unchanged. The reported findings regarding the linear physics of metamaterials, while important and carried out carefully, do not go beyond previous work. I acknowledge that the compact footprint and low disorder of the resonator studied here holds great potential for scalability. However, I do not believe the manuscript makes a compelling case for demonstrating sufficient control over the multi-mode response of the systems at the same level as conventional superconducting resonators. To elaborate further, while the measured disorder in the metamaterial structure is low, it is unclear how precisely aligned measured devices are to original designs. The authors mention that run-to-run variability in kinetic inductance introduces up to 15% change in device frequencies, which is significantly larger than conventional devices based on geometric inductance. It seems that while disorder within a single chip is low, there are random factors that may affect the run-to-run reproducibility of multiple chips made using the same procedure—potentially due to variations in the kinetic inductance, which is sensitive to wire dimensions, film thickness and resistivity. A thorough analysis needs to account for these factors, for example by comparing the absolute values of design parameters vs measurements across multiple fabrication cycles. Alternatively, the authors could provide a clear case of an operational linear system with a degree of complexity comparable to those obtained via conventional techniques (such as the impedance matching 'tapering' circuit in of Ref. 24).

Arguments for the increased light-matter interaction and applicability to giant atoms are compelling. But again, without reproducible control of parameters like resonator frequency between fabrication cycles, the ultimate feasibility of this

approach remains unknown. Therefore, I remain on the opinion that this work is better suited for a more specialized journal. A final small suggestion (relating to questions #5 and #6 from my initial review) is to add a clarifying statement in Section D that the disorder extracted from the simulations/measurements is sufficient to explain the trend of Δ -topo in figure 3d, f. After reviewing this section and figure 4, it was still not obvious to me that this was the case.

Reviewer #4

(Remarks to the Author)

REVIEWER COMMENTS

Reviewer #1 (Remarks to the Author):

The authors investigate the use of high kinetic inductance materials for the fabrication of coupled cavity arrays (CCAs). CCAs have been proposed as platforms for quantum simulation as cavity frequencies and couplings can be systematically varied. Early work with CCAs [refs. 15, 18] examined large 2D arrays, and while the research was motivated by interesting physics, the experimental results were rather ambiguous. The present manuscript by Jouanny et al. takes a needed step back and carefully examines 1-dimensional (1d) cavity arrays fabricated with NbN, a high kinetic inductance material.

The research team first characterizes CCAs with one cavity per unit cell. With the exception of some slot line modes at high frequencies, the cavity transmission spectrum is consistent with expectations. They then investigate CCAs with more than one cavity per unit cell. These measurements show the expected bands and bandgaps. Lastly, CCAs with staggered coupling rates are fabricated to investigate the SSH model. The expected topological modes (and their hybridization) are observed and characterized using a novel time-domain measurement. A major outcome of the manuscript is the characterization of the disorder, which is about 11 MHz, as compared with the cavity frequency ~ 5 GHz.

Obviously, a large collection of coupled harmonic oscillators is going to exhibit a complex normal mode spectrum. The present manuscript goes beyond refs. 15 and 18 in that it demonstrates a deep understanding of the measured normal mode spectrum. As such, I strongly encourage publication in Nature Communications after the authors address a few questions.

We thank the Referee for their positive feedback on our work.

- The authors refer to a hexagonal geometry (Figs. 1 c,d), but upon examining the device architecture, it appears to just be a 1d CCA that is patterned in a zig-zag pattern. Specifically, it only consists of a 1d array of resonators, with each resonator coupled by an inter-digitated capacitor. It could of course be extended to a larger 2D hex array, but this is not demonstrated in the manuscript. As the results from the "hex" geometry are not central to the paper, I encourage the authors to more accurately describe the geometry in Figs. 1(c,d), or remove these panels. As is, the results are somewhat overstated.

We thank the Referee for this insightful comment. We understand that referring to this additional design as "hexagonal" could cause confusion with a 2D array. In the revised manuscript, we clarify that this structure is, in fact, one-dimensional (1D). We chose to describe the CCA as "hexagonal" (or "rectangular") to refer specifically to the shape of the unit cell rather than the geometry of the overall lattice. Presenting this 1D hexagonal CCA in the first figure, we believe, gives the reader a clearer understanding of the platform's versatility. This 1D hexagonal CCA configuration allows us to increase the intercell coupling J while maintaining control over the total cell capacitance, facilitating a transition to a left-handed coupled cavity array. Additionally, extending this design to a 2D structure would require further adjustments to the unit cell, even for the current hexagonal CCAs. To avoid any potential confusion, we have adopted the Referee's suggestion to refer to this alternative CCA configuration as a "zigzag-design".

In the new version, we made sure to clarify this point with the following sentence in the main text:

"We design, simulate, fabricate, and investigate 1-dimensional (1D) CCAs comprising rectangular-shaped (Fig.1a and b) and hexagonal-shaped base units (Fig.1c and d)."

And the following the sentences in the caption of Figure 1:

“a. Optical micrograph of a representative linear-design CCA comprising 25 resonators with rectangular base unit.”

“c. Optical micrograph of a representative zigzag-design CCA comprising 26 resonators with hexagonal base unit.”

- The authors refer to “ghost ports” near the end of section 1A. While the need to engineer parasitic capacitances to maintain uniform cavity frequencies is apparent, the manner in which this is achieved is unclear from the text and Fig. 1b. I therefore appreciate it if the authors can comment in more detail on how parasitics at the ends of the CCA are managed.

We thank the Referee for this valuable remark. At the edges of the CCA, chiral symmetry is naturally broken due to a differing total capacitance, which shifts the resonant frequency of the edge cavities. There are two primary methods to maintain symmetry:

1) Adjust the inductor length of the edge cavities so that they resonate at the same frequency as the other cavities. However, this approach alters the impedance at the edges of the CCA chain.

2) Modify the capacitive environment at the edges to match the capacitance of the edge cavities with that of the interior cavities. A straightforward method is to introduce a microwave port in the form of a resonator without grounding the inductor. This approach requires minimal redesign, provides automatic capacitance matching, and is the method we have adopted in this work.

Following the Referee’s comment, we modified the sentence in the main text introducing the ghost ports to make it clearer:

“To ensure this degeneracy, we introduce driving ports shaped and positioned exactly as a cavity of the array but whose inductor is not shunted to ground. Such ghost ports ensure a uniform capacitive environment for both edge and bulk resonators (see blue false-colored part in Fig.1 b).”

We also added the following sentence in the description of the figure:

“Notice that the coupling port (blue) does not have an inductor shunted to ground.”

- Minor: a) Consider removing Figs. 1c,d and consolidating Fig. 1. b) There appears to be an inconsistency between the legend of Fig. 1c, where $J/2\pi = 164$ MHz is listed, and the caption, where $J/2\pi = 200$ MHz is listed. c) The formatting of the references is somewhat muddled, e.g. 2, 4, 5, 22, 30, ... The references should be cleaned up before resubmission.

We thank the Referee for their keen observation:

- a) We believe that showing this CCA with hexagonal base unit in the first figure provides the reader a better grasp of the versatility of our platform, and respectfully prefer to keep panels c and d. But, as reported in our reply to one of the previous questions, following the Referee’s suggestion we now refer to this alternative CCA configuration as a "zigzag-design."
- b) Caption of Fig.2 has been corrected to $J/2\pi = 164$ MHz.

c) The citations have been fixed.

Reviewer #2 (Remarks to the Author):

The manuscript by Jouanny et al (NCOMMS-24-22592-T) presents a new experimental circuit-QED architecture for implementing large-size photonic lattices described by tight-binding models (and potentially able to be coupled to artificial atoms). The setup consists of an array of capacitively coupled resonators made of high kinetic inductance NbN thin film, which compared to state-of-the-art devices allow for a significant size reduction, so that the authors can assemble up to 100 resonators within a few mm² chip. This is achieved by keeping the amount of disorder comparable with that of larger devices.

The authors in particular realise and test (mostly through transmission spectra):

- multi-band lattices (up to five bands) featuring multiple bandgaps;
- the SSH model in the topological non-trivial phase;
- edge modes occurring in the SSH model when this is in a non-trivial topological phase for a varying number of resonators in the array.

Edge modes, and especially their frequency splitting, are studied as a function of disorder and this is proposed as an interesting method to measure the disorder amount of the entire array relying on the bulk-edge correspondence principle.

I think the work is a high-quality contribution to the concerned literature, marking a milestone that has the potential to spur further advancements in the field, both of experimental and theoretical nature. I particularly appreciated the presentation, which is very clear and accurate (including the three figures in the main text). For these reasons, I support publication on Nat Comm only provided that the following points are addressed by the authors.

We thank the Referee for their positive assessment of our work.

1) Given its importance in the present work, I believe it is useful for the reader to recall explicitly the definition of chiral symmetry and the definition of the chiral symmetry operator for the SSH model;

We thank the Referee for this remark and agree that it is important to recall with more details the definition of chiral symmetry.

We modified section C in the main text to include the definition of chiral symmetry.

After equation (3) of the main text, we added:

“While periodic boundary conditions were applied to derive the Hamiltonian in Eq. (3), to observe SSH edge states, the CCA must have open boundary conditions. Furthermore, the mid-gap edge states are well defined only if chiral symmetry is realized, i.e., when $\hat{\Gamma} \hat{H} \hat{\Gamma}^\dagger = -\hat{H}$ holds, where $\hat{\Gamma}$ is the chiral symmetry operator. Specifically, for the SSH model, $\hat{\Gamma}$ is defined using the sublattice projectors associated to the A and B sites, \hat{P}_A and \hat{P}_B , as $\hat{\Gamma} = \hat{P}_A - \hat{P}_B$ [38,59].”

2) Among platforms beyond circuit QED for investigating atoms/emitters coupled to photonic lattices, the Introduction mentions Ref. [10] dealing with atoms in photonic crystals. It is probably fair mentioning as well cold atoms in optical lattices, see e.g. Krinner et al, Nature 559, 589 (2018);

We thank the Referee for bringing this to our attention. We added the indicated reference in the introduction.

3) Ref. [66] does not deal with “giant-atom photon bound states”, while these are indeed featured in Ref. [67] but only for the simplest special case of a homogeneous (one-band) lattice. The general theory of these states for an arbitrary photonic lattice (which complies very well with the spirit and outlook of the present manuscript) was recently presented in L. Leonforte et al., arXiv:2402.10275.

We thank the Referee for pointing this out. Indeed, Ref. [66] of the original submission of the manuscript does not include giant-atom photon bound states, but giant atoms in standard waveguide with linear dispersion. We corrected this in the main text and added the suggested reference.

Some very minor suggested corrections on the text follow.

- page 3: “arise from two origins”->“have a double origin”;
- end of column 1, page 3, add “quality factor” before Q; shortly after “doesn’t”->“does not”;
- page 4, column 1: the sentence “The SSH chain is a ubiquitous chiral symmetry protected topological phase of matter that manifest in 1D systems” is a bit awkward (a chain is not a phase; also I do not see what ubiquitous means). I would just replace it with “The SSH chain can be in/can host a chiral-symmetry-protected topological phase”;
- to comply better with conventional terminology, I would recommend using “edge modes” (or “edge states”, even though “modes” is more correct), instead of “boundary modes”. Somewhat similarly, I would recommend replacing “SSH states” with “SSH edge states” (or “modes”) which is still a rather short name.

We thank the Referee for the suggestions. All the corrections have been implemented in the manuscript.

Reviewer #3 (Remarks to the Author):

In this paper, the authors characterize high-kinetic-inductance coupled cavity arrays using NbN thin films, achieving a low footprint, precise control over circuit parameters, and low-frequency disorder. They fabricate devices with up to 100 cavities in different lattice configurations to study devices with multiple passbands. They then quantify the level of disorder in cavity arrays using the hybridization between edge modes of the topologically non-trivial SSH chain. Overall, the paper is interesting and well-written, with suitable physical intuition provided for understanding these large multimode systems. The work supports the main claims of the text, and the results are communicated effectively.

In my estimation, the main advancements of this work are (1) improved parameter control and miniaturization of superconducting metamaterials and (2) a novel method of quantifying disorder in the bulk. These improvements, while important and useful, are highly technical. The Su-Schrieffer-Heeger lattice has been studied previously using superconducting resonators, and multiple previous works have demonstrated coupling quantum emitters to coupled-cavity-arrays, even investigating qubit-qubit interactions in this setting. Because this work is mainly focused on technical improvements and the linear physics of metamaterials, I believe a more specialized journal would be more suitable.

We thank the Referee for their overall positive assessment of our work, though we respectfully disagree with their view that our progress is solely technical. We believe our work has broader implications beyond technical advancements, as we elaborate below. In particular, the progress we have achieved in metamaterial engineering is likely to drive developments across multiple fields, as also noted by the other referees. Indeed, what may initially be perceived as a technical improvement often plays a critical role in unlocking the full potential of novel platforms.

Regarding point (1) mentioned by the referee, achieving miniaturization while maintaining low disorder is a very important and not at all trivial technical achievement. Such progress in material and device engineering opens up vast opportunities for future research, and is at the heart of how profound scientific progress is made. To further illustrate the impact of our platform, we have provided a comparison (figure on the right) between standard CPW technology [Göppl, AIP 2008], lumped-element resonators fabricated with traditional superconducting platforms [Kim, PRX 2019], and our new NbN-based high-kinetic-inductance ultracompact platform.

From this comparison, it becomes clear that, on a typical $1\text{ cm} \times 1\text{ cm}$ chip, we can now accommodate several hundred NbN resonators. **This drastically increases the ability to define complex and dense bosonic densities of states, without compromising quality or increasing scattering (disorder).**

Just to mention a few of the novel opportunities that our compact CCA platform enables:

[figure redacted]

Fig. 1: Comparison between (a) a $50\ \Omega$ coplanar waveguide (CPW) distributed resonator [Göppl, AIP 2008], a lumped element resonator based on (b) a standard superconducting film [Kim, PRX 2019], and (c) a high kinetic inductance thin film, realized in our HQC lab.

- Exploration of complex and high density of states:** The low disorder in our ultracompact CCA platform enables the detailed study of complex densities of states that are not easily attainable with alternative technologies. The long-term goal is to implement non-quadratic Hamiltonians with large systems that cannot be efficiently simulated using conventional computers. The fundamental building blocks of such architectures are: (a) A large number of linear modes that can be coupled together. (b) One or multiple quantum emitters that interact with these modes.

Regarding point (a), not all approaches are viable; we argue that miniaturization, while maintaining control over coupling rates, is essential to achieve a density of states with sufficiently fine discretization, given the finite number of elements. Our CCA platform is easily extensible to planar arrays, with 100x100 resonators CCA fitting into a 5x5 mm² chip. Suppression of unwanted chip modes on a large area requires careful engineering of ground vias and can additionally be impeded by the finite size effect of light propagation. In addition, in systems with higher complexity, such as those exhibiting multigap spectra, scattering in resonator parameters can easily obscure important spectral features, as we demonstrate in the paper for the finite splitting of the edge modes. As for point (b), maintaining strong coupling between the linear modes and the quantum emitter is crucial. Therefore, we argue that using compact but highly lossy elements, such as parallel plate lumped resonators operating at low powers, is not suitable for these experiments. We believe our platform—with its low disorder, ultracompact CCA, and minimal loss—enables the detailed study of complex densities of states that are otherwise unattainable with alternative technologies. This paves the way for analog quantum simulations of flat bands, vibronic phenomena, and spin-boson Hamiltonians (see Kim, Phys. Rev. X Quantum).
- Boosting light-matter interaction:** Our compact high-kinetic-inductance resonators, with reduced self-capacitance, result in higher resonator impedances (exceeding 1 k Ω). Because the electric coupling strength with artificial atoms scales with $\sqrt{Z_r}$, this design enables a significant enhancement of light-matter interactions [39], even reaching the regime of multimode ultrastrong coupling, where many-body physics can emerge [45]. In particular, this feature is essential for the bosonic environment to strengthen electrical coupling to alternative effective two-level systems with small electric dipole moments, such as charge and spin systems in semiconducting quantum dots [40].
- Facilitating interactions with “giant atoms”:** Our ultra-compact CCA naturally supports multiple coupling points with standard superconducting qubits. This enables the realization of discrete waveguides coupled to giant atoms, a concept that has so far been explored only theoretically in recent works [69,70]. With distributed or standard LC CCAs, defining coupling to multiple points with a transmon qubit will require bending of the metamaterial in order to increase the electrical length and accumulate enough phase delay between multiple coupling points (see Ferreira, Phys. Rev. X). Preserving low disorder while bending the metamaterial is very challenging.

With respect to point (2) mentioned by the referee, as the Referee has already noted, our method for quantifying and studying scattering in CCA systems is not only novel but also highly general. By leveraging topological concepts—specifically, the bulk-boundary correspondence—we theoretically proposed and experimentally explored a new tool for testing the disorder in the CCA, which we refer to as a “*topological disorder-meter*”. This is an innovative approach that moves beyond the conventional method of studying the disorder based on simply identifying all the bulk modes and comparing them with simulations [16]. Thus, we reveal a useful application for topology which allows one to overcome the need for direct access into the bulk when analysing the quality of a metamaterial structure. This is especially useful in the study of cavity arrays, which are commonly probed via lateral input-output probes. Furthermore, this method is not limited to the superconducting community; it

applies to any SSH-type (and more general symmetry-protected topological) metamaterial, and facilitates efficient study and design of more sophisticated lattices where degeneracies are broken by disorder.

In conclusion, we are confident that the combination of our technological advancements and the novel conceptual application of topology will appeal to the broad readership of *Nature Communications*. Our results are not only of technical interest but also open up new research directions, both experimentally and theoretically.

In order to properly emphasize the novel aspects and potential of the introduced ultracompact CCA platform, we have added the following text:

After the following section:

Notably, the high kinetic inductance of our devices enables the realization of high-impedance resonators in the array. This characteristic increases the coupling to the charge degree of freedom of both superconducting [38] and semiconducting [39] qubits, enhancing the possibility of achieving the ultra-strong coupling regime [40].

Added:

This ultracompact, high-impedance platform significantly enhances the density of modes within a given footprint, without compromising the quality or increasing the disorder. This enables the exploration of complex and high density of photonic states necessary for emulating the effect of highly structured non-Markovian environment of open quantum system [17, 18].

After the following section:

On the other hand, photon lattices also offer promising avenues for future experiments aimed at investigating quantum phase transitions [1, 71]. By effectively reducing random disorder, cavity arrays can be fabricated with controlled levels of disorder, potentially enabling the study of many-body localization effects [46]. While our current work remains non-interacting, the incorporation of interactions is feasible through the inherent nonlinearity present in high kinetic inductance materials, resulting in both χ^2 and χ^3 nonlinearities [72], or by integrating qubits into each resonator [71]. We aim to leverage nonlinearities in CCAs as a novel tool for studying driven-dissipative phase transition [71, 73, 74]. Moreover, these lattices facilitate the creation of unique devices capable of hosting photons in curved spaces [19], gapped flat band [75,76], and novel forms of qubit-qubit interaction [20, 76].

Added:

The high-impedance, ultracompact nature of our platform opens new frontiers for realizing dense, multimode environments [7,8,42], potentially enabling the study of quantum many-body phenomena with a high degree of control. Its scalability, without sacrificing quality or increasing disorder, paves the way for future explorations of light-matter interactions in unconventional regimes, such as multimode ultra-strong coupling [9,45,66] and complex photonic states for analog quantum simulation [17,18,24]. Additionally, our topological disorder-meter serves as a novel and versatile tool for characterizing disorder in a wide range of quantum or classical systems, and is, in principle, applicable to any symmetry-protected topological phase, such as Kitaev chains [78,79] or 2D honeycomb lattices [80]. These improvements position our platform as a flexible foundation for advancing quantum technologies [2,3] and specifically analog quantum simulation [17, 18, 24].

Below, I have attached some comments and questions.

1. Can the authors comment on the origin of chip slot modes? Does this refer to slot-line waveguide modes (as discussed in <https://pubs.aip.org/aip/apl/article/104/5/052602/828862/Fabrication-and-characterization-of-aluminum>) or packaging modes? In the former case, does the grounding wire between resonators (in the rectangular design) fail to suppress these modes?

We thank the Referee for raising this question. We attribute these chip slot modes to slot-line waveguide modes, which are enhanced by the dense layout of our chip, typically containing three devices on a 4x7 mm² chip. Although grounding wires are implemented, they do not fully suppress these modes. We believe this effect arises from the high kinetic inductance of the films, as the grounding strips between the resonators typically exhibit an inductance of around 3 nH. This prevents them from providing an ideal connection between the two ground planes surrounding the CCAs, thereby enabling the presence of slot-line waveguide modes.

To mitigate these effects, we incorporate dense bridge wire bonds around each CCAs.

In the figure below, we report transmission in the coupled cavity array with 100 cavities shown in the paper (Fig.2b of the main text). The blue data set is the dataset shown in the paper. The orange dataset is an earlier dataset of the same device but without bridge wirebonds around the CCA.

Between these two datasets we can observe two main differences:

1. The amplitude of the slot modes is higher in the orange dataset compared to the blue dataset.
2. The frequency of the slot modes is shifted upwards in the blue dataset. This frequency shift is likely due to the reduced inductance between the ground planes around the CCA.

Together, these observations indicate that the chip is indeed exhibiting slot-line waveguide modes. We believe that these modes could be further suppressed by using denser airbridges or a ground plane made from a standard superconductor, such as aluminium.

2. Can the authors comment on whether the large kinetic inductance fraction of the resonators results in a strong sensitivity in the resonator frequency to temperature due to the thermal population of quasi-particles? Does this contribute to the disorder?

We thank the Referee for raising this insightful question. The following answer is based on our previous work (Ref. [36] of the main text).

Due to the high critical temperature of NbN ($T_C \approx 7.1K$), in comparison with Al (see Fig. 2 of Ref. [36]), the superconducting gap, as approximated by BCS theory ($\Delta \approx 1.764k_B T_C$), is relatively large. This sizable superconducting gap provides enhanced protection against thermal quasiparticles. While we did not measure the frequency shift versus temperature in this study, our previous work (see Fig. 6 of Ref. [36]) on various NbN film compositions using resonators of similar geometry demonstrates this effect. Specifically, for films with comparable properties ($T_C \approx 7.1 K$), the influence of quasiparticles remains minimal at temperatures below 600 mK, with the resonator frequency shifting by only -50 kHz at 600 mK. Given the moderate change in kinetic inductance and T_C between the two studies, we expect a similar behavior from this film. Below this temperature, Two-Level System (TLS) fluctuators become the dominant loss mechanism, though their effect is also minimal—causing a frequency shift of up to +50 kHz at 600 mK.

Moreover, from what is shown in Ref. [36], it seems that these two effects tend to counteract each other, resulting in a minimal net frequency shift with temperature below 600mK. Quantitatively, the frequency shift due to combined quasiparticle and TLS effects is estimated to be $\frac{\sigma_z^{QP+TLS}}{f_r} \approx 4 \times 10^{-4}\%$, where $\sigma_z^{QP+TLS} = 20\text{kHz}$ is the maximum frequency shift caused by the combined effect of quasiparticles and TLS below 200mK, and f_r is the resonant frequency, taken as 5 GHz (as in the main text).

At our operating temperature of 10mK, this shift will even be smaller and well within the error margin of our disorder estimation. Hence, it has a negligible impact on the CCA spectrum.

3. When discussing adding multiple resonators/unit cells, the term “resonant condition” is used, which refers to realizing resonators with degenerate frequencies in the entire chain. I believe the wording can be made clearer here to get this point across – an initial read led me to think this was a more involved requirement.

We thank the reviewer for this helpful suggestion. To clarify, in contrast to mono- or di-merized chains, where chiral symmetry is automatically satisfied, achieving resonant frequency degeneracy across multiple cavities per unit cell (three or more) requires fine-tuning of the inductor length.

The calibration process is explained in Appendix C, Fig. 12. It involves more than just adjusting the inductor length to match a target frequency. Changing the inductor length also slightly modifies the cavity capacitance, resulting in a frequency shift. Therefore, we had to iterate finite element simulations to finely tune the design and maintain constant resonant frequencies across the cavities.

We have revised the wording in the manuscript to make this point clearer:

“However, for $M > 2$, the cavities do not necessarily have the same total capacitance and hence maintaining a constant resonant frequency for all cavities requires fine tuning of the inductors to compensate for this effect [44].”

4. Is there a typo in the second term of Eqn. (3)? This term is not Hermitian. I believe it should be something like $(J_1 + J_2 * \cos(kd)) * T_x + J_2 * \sin(kd) * T_y$. A related comment is that this Hamiltonian, strictly speaking, refers to the SSH chain when periodic boundary conditions are applied. In other

words, in the absence of disorder and next-nearest-neighbor hopping, there is still no zero-mode (edge mode) in the spectrum of this Hamiltonian. I think some clarifying statements on this would be helpful for readers.

We thank the Referee for pointing this out. The Referee is correct, there is indeed a typo in the second term of Eqn. (3), the Hermitian conjugate was missing. We corrected the Hamiltonian in the main text.

Regarding the second point, we agree with the referee's observation. To clarify this for the readers, we have added the following statement to the manuscript:

"While periodic boundary conditions were applied to derive this Hamiltonian, to observe SSH edge states, the CCA must have open boundary conditions."

5. Based on Eqn. (4), we expect the splitting between the edge modes to reduce with larger N. This is true for the weakly localized chain ($J_2/J_1 = 1.22$) but does not appear to be true for the strongly localized chain ($J_2/J_1 = 1.57$) in Fig. 3(d, f). Can the authors comment on what is happening?

We thank the Referee for raising this important point. We recognize that this aspect could have been made clearer in the manuscript. The Referee is right: in an ideal scenario without disorder, we expect the splitting between the SSH modes to decay exponentially with the number of cavities, as indicated in Eqn. (4). This effect motivated our investigation into the impact of disorder on SSH modes and its implications for our arrays.

The effect the Referee mention is shown in Fig.4a where we report the splitting between the SSH modes, Δ_{Topo} , as a function of the number of cavities in the case $J_2/J_1 = 1.22$. Here, one can clearly see that, as the Referee observed, the data do not agree with the ideal scenario of Eqn. (4), and for instance $\Delta_{Topo}^{32} < \Delta_{Topo}^{64}$. But this feature can be explained including disorder in the simulation as explained in part "D. Disorder" in the main text.

To make this clearer, we added the following sentence to the manuscript at the end of part "C. Engineering localized modes" in the main text.

"However, as shown in the right panels of Fig.3d and f, the measured splitting between the SSH edge modes does not always decrease as a function of N, contrarily to what is predicted by Eq. (4). In particular, for the specific realization of the $J_2/J_1 = 1.57$ case (Fig. 3f), the splitting Δ_{Topo} , even appears to increase with N. As discussed in the next section, this effect is attributed to disorder."

6. We also expect that the edge-mode splitting should be reduced when the localization increases, based on Eqn. (5). However, in Fig. 3(d, f), this does not appear to be the case in the N = 32 and N = 64 cases. Can the authors comment on what is happening?

We believe we addressed this point in the answer to the previous question.

7. Can the authors comment on why the edge mode splitting can be fully explained by disorder in the frequency (τ_z)? Can they rule out the possibility that other effects (beyond the SSH model) might play a role? For example, could coupling to a slot-line mode or packaging mode result in any hybridization? The role of Tx disorder is quite minimal, as shown in Fig. 14. What about disorder in J' (next-nearest-neighbor coupling, appearing in T_0)?

We thank the Referee for this insightful question. Experimentally, outside of the Hamiltonian, apart from potential slot-line modes, we have not identified other factors that could significantly affect the coupled cavity array (CCA) spectra. Coupling to slot-line mode, would indeed shift the SSH modes. However, for all the SSH devices fabricated we did not discern any slot modes in the bandgaps. All the spectra for the different batches are shown in Fig.3 of the main text and Fig.19 of the supplementary information.

From the Hamiltonian perspective, we can have noise on the prefactors of τ_0 , τ_z and τ_x/τ_y . Beside the noise in τ_z discussed in the main manuscript, we also show, in Fig.14 of the supplementary material, the resilience of Δ_{Toppo} against disorder on τ_x (also valid for τ_y). We clearly observe that those other noise channels lead to minimal frequency scattering.

This leaves us with errors along τ_0 due to next-nearest neighbor coupling terms (J'). Below is a simulation performed similarly to Fig. 14 of the supplementary material. In this case, we apply disorder on the next-nearest neighbor coupling (J'), which results in equivalent disorder along τ_z . We denote the noise proportional to J' as σ' .

First, we increase σ' , while keeping σ_x and σ_z set to 0 (Panel **a** of the figure below). As for the disorder along τ_x in Fig.14 b of the supplementary material, we see a minimal influence of σ' on Δ_{Toppo} compared to the resonant frequency disorder. In panel **b**, we replicate the simulation from Fig.14 **c** of the supplementary material, where we increase σ' for the same values as in panel **a** but with constant resonant frequency disorder of $\frac{\sigma_z}{f_r} = 0.2\%$. Here the influence of σ' is overshadowed by the resonant frequency disorder.

To conclude, we believe that this source of noise of negligible in our analysis.

We added the following figure (Figure 2) as well as the following text in the supplementary material.

“In Fig.15 we implement a similar study as in Fig.14, with disorder on the next-nearest neighbor coupling, J' , namely, σ' . With J' being proportional to τ_0 (Eq. (3) of the main text) σ' -disorder will result in σ_z -type disorder. In Fig. 15a, as in Fig. 14b, we observe a considerable deviation just for high enough values of σ' and N . In Fig. 15b, we apply the same disorders as in Fig.15a but with constant resonant frequency disorder $\sigma_z / f_r = 0.2\%$, as in Fig. 14c. Again, here one can see that resonant frequency disorder, σ_z , dominates over the next-nearest neighbor coupling disorder, σ' .”

Figure 2: Simulation of the influence of σ' disorder on the SSH edge mode splittings. The simulations are performed with a Hamiltonian on a CCA in the SSH configuration with $J_2/J_1 = 1.22$. **a.** Splittings of the SSH edge modes, Δ_{Top0} , as a function of the number of resonators, N , for different values of σ' disorder ($\sigma_x = 0$ and $\sigma_z = 0$). **b.** Δ_{Top0} as a function of N for the same values of σ' disorder as in panel **a** ($\sigma_z = 0.2\% f_r$).

Reviewer #4 (Remarks to the Author):

Reviewer #1 (Remarks to the Author):

The authors have modified the manuscript in response to suggestions from referees 1 and 2. I tend to disagree with referee 3, as the manuscript is one of the strongest "cavity lattice" papers published to date. As such, I recommend publication as is in Nature Communications.

We appreciate the positive feedback and recognition from the referee.

Reviewer #2 (Remarks to the Author):

The authors satisfactorily addressed all of my concerns. I therefore recommend the manuscript for publication.

We thank the referee for their positive assessment of our work.

Reviewer #3 (Remarks to the Author):

I appreciate the authors' efforts to address my comments. I am satisfied with the answers to my technical questions.

However, my assessment of the scope impact of this manuscript remains unchanged. The reported findings regarding the linear physics of metamaterials, while important and carried out carefully, do not go beyond previous work. I acknowledge that the compact footprint and low disorder of the resonator studied here holds great potential for scalability. However, I do not believe the manuscript makes a compelling case for demonstrating sufficient control over the multi-mode response of the systems at the same level as conventional superconducting resonators. To elaborate further, while the measured disorder in the metamaterial structure is low, it is unclear how precisely aligned measured devices are to original designs. The authors mention that run-to-run variability in kinetic inductance introduces up to 15% change in device frequencies, which is significantly larger than conventional devices based on geometric inductance. It seems that while disorder within a single chip is low, there are random factors that may affect the run-to-run reproducibility of multiple chips made using the same procedure—potentially due to variations in the kinetic inductance, which is sensitive to wire dimensions, film thickness and resistivity. A thorough analysis needs to account for these factors, for example by comparing the absolute values of design parameters vs measurements across multiple fabrication cycles. Alternatively, the authors could provide a clear case of an operational linear system with a degree of complexity comparable to those obtained via conventional techniques (such as the impedance matching 'tapering' circuit in of Ref. 24).

Arguments for the increased light-matter interaction and applicability to giant atoms are compelling. But again, without reproducible control of parameters like resonator frequency between fabrication cycles, the ultimate feasibility of this approach remains unknown. Therefore, I remain on the opinion that this work is better suited for a more specialized journal.

We thank the referee for highlighting this important point. As noted in the manuscript, we previously observed up to 15% variability in resonance, leading to approx. 32% variability in the sheet kinetic inductance across fabrication batches between different deposition runs. However, it is important to emphasize that this value represents the tail of the distribution and was observed only once. This level

of variability occurred in cases where the sputtering chamber had been opened, and the Nb target dismounted and re-mounted, which historically caused significant run-to-run variations when the same recipe was repeated. It is possible that in this specific case the target was mounted incorrectly, as this variation, now that we have enough statistical evidence, has never happened again. To address this issue, we implemented a monitoring step where the room-temperature normal-state sheet resistance of each wafer is measured after the first batch deposition, particularly following any venting of the sputtering system.

As expected, we identified that variations in kinetic inductance were strongly correlated with the normal-state resistance, suggesting slight difference in deposition rate depending on the target installation. Using this insight, we introduced minor adjustments to the deposition time for each recipe based on the observed sheet resistance. Since implementing this monitoring and adjustment protocol six months ago, we have observed a significant improvement in reproducibility, with the standard deviation of sheet kinetic inductance across depositions dropping to approximately 3%. This 32% variation chip was measured and included in the paper; we felt it was important to report this extreme value for the sake of completeness, even though it was an outlier. It is also important to note that the films exhibit remarkable uniformity, as demonstrated by the low disorder we have characterized. Furthermore, once the film is calibrated on a wafer scale, the design can be adjusted to account for any variations in kinetic inductance.

In light of recent improvements, we have added the following statement in the methods section. We highlight in blue below what has been added:

“We observed changes in L_k , \square due to potential fluctuations in the film deposition process (thickness or composition), resulting in up to 15% change of the CCA resonant frequency, $\omega_r/2\pi$. This variation of kinetic inductance has been reduced to $\sim 3\%$ with improved monitoring and deposition adjustments.”

Moreover, we agree with the referee that the control we have on high-kinetic inductance superconducting thin films will never be as good as the control one can get on devices where inductance is obtained by purely geometrical effects, because of the presence of another variable. However, on par with the referee’s argument, Josephson junctions— while being the foundational building blocks of superconducting quantum computers—exhibit similar batch-to-batch inductance variability (around 10% even in a well-controlled industrial setting <https://arxiv.org/abs/2403.01312>). On a single wafer the variability can be reduced to $\sim 4\%$ for standard process or even $\sim 2\%$ within a long oxidation time, as reported in Phys. Rev. Research 5, 043001 (2023) . Moreover, our current 15% variability, is observed in a general-purpose academic sputter system, and it can be significantly improved within a more dedicated industrial setting.

As for implementing more advanced designs (such as the tapering circuit mentioned by the referee), we agree that this is an intriguing direction for realizing a compact slow-light waveguide. However, exploring such a design would require additional fabrication runs, which extend beyond the scope of the current study. We plan to investigate these possibilities in future work, as they could further demonstrate the versatility and scalability of our platform.

A final small suggestion (relating to questions #5 and #6 from my initial review) is to add a clarifying statement in Section D that the disorder extracted from the simulations/measurements is sufficient to explain the trend of Δ -topo in figure 3d, f. After reviewing this section and figure 4, it was still not obvious to me that this was the case.

The referee is correct, this can be further clarified in the text. We added a clarifying statement in the text. We highlight in blue below what has been modified.

“To investigate the impact of disorder on the SSH edge modes, we fabricate and characterize additional SSH devices [44] with a coupling ratio of $J_2/J_1 = 1.22$, as illustrated in Fig. 3d. In Fig. 4a, we present a dataset comprising ΔN_{Topo} values extracted from the measured CCAs as a function of chain length N . Notably, while ΔN_{Topo} exhibits the expected exponential decay with respect to N , it does not asymptotically approach zero, demonstrating significant deviations from the theoretical prediction (black line ~~vs red~~ in Fig. 4a).

To rigorously account for this observation, we conduct numerical simulations that introduce Gaussian noise, denoted as σ_L , applied to the inductance values of all resonators within the chain (red line in Fig. 4a). The choice to introduce scattering in L_g as the main noise source is motivated by the fact that the inductors have the most critical dimension in the resonator design, rendering them more susceptible to scattering during the fabrication process. The σ_L noise applied to the inductors induces both τ_z and τ_x type of disorder in Eq. (3), impacting respectively the resonant frequency and the coupling of the resonators in the CCA. Our analysis primarily focuses on the principal effect of σ_L scattering namely τ_z -type disorder (breaking chiral symmetry), which we refer to as $\sigma_L \rightarrow z$. These simulations accurately describe the behavior of ΔN_{Topo} observed in Fig. 3d and f.”

Reviewer #4 (Remarks to the Author):
